# Fish Waste: From Problem to Valuable Resource

**DOI:** 10.3390/md19020116

**Published:** 2021-02-19

**Authors:** Daniela Coppola, Chiara Lauritano, Fortunato Palma Esposito, Gennaro Riccio, Carmen Rizzo, Donatella de Pascale

**Affiliations:** 1Department of Marine Biotechnology, Stazione Zoologica Anton Dohrn, Villa Comunale, 80121 Naples, Italy; daniela.coppola@szn.it (D.C.); chiara.lauritano@szn.it (C.L.); fortunato.palmaesposito@szn.it (F.P.E.); gennaro.riccio@szn.it (G.R.); carmen.rizzo@szn.it (C.R.); 2Institute of Biochemistry and Cell Biology (IBBC), National Research Council, Via Pietro Castellino 111, 80131 Naples, Italy

**Keywords:** fish waste, fish byproduct valorization, marine sustainable sources, collagen, bioactive peptides, chitin, oil, enzymes

## Abstract

Following the growth of the global population and the subsequent rapid increase in urbanization and industrialization, the fisheries and aquaculture production has seen a massive increase driven mainly by the development of fishing technologies. Accordingly, a remarkable increase in the amount of fish waste has been produced around the world; it has been estimated that about two-thirds of the total amount of fish is discarded as waste, creating huge economic and environmental concerns. For this reason, the disposal and recycling of these wastes has become a key issue to be resolved. With the growing attention of the circular economy, the exploitation of underused or discarded marine material can represent a sustainable strategy for the realization of a circular bioeconomy, with the production of materials with high added value. In this study, we underline the enormous role that fish waste can have in the socio-economic sector. This review presents the different compounds with high commercial value obtained by fish byproducts, including collagen, enzymes, and bioactive peptides, and lists their possible applications in different fields.

## 1. Introduction

Global use of natural resources has been increasing substantially in recent years, reaching 92.1 billion tons in 2017 and resulting in an increase of 254% from 27 billion in 1970, with the rapid build-up in the annual extraction since 2000 [1]. Because of this, the research is improving its efforts in building a circular bioeconomy, which aims to enhance the value of material flows and to achieve sustainable consumption and production. It represents an emerging concept that elicits great attention for the purpose of efficient and sustainable use of resources, energy and infrastructure to ensure the quality of life of humans. In this framework, biomass wastes play a major role in the implementation of circular bioeconomy, based on the reuse and recycling of materials to reduce waste production.

In the last years, fish products consumption has seen a huge increase following its recognition as a key component of a balanced diet and healthy lifestyle. Based on the Food and Agricultural Organization (FAO), United States, the total fisheries and aquaculture production has shown a considerable increase of more than eightfold between 1954 and 2014, driven by advances in fishing technologies and rapid developments in aquaculture; in 2014, the global fishery production was 93.4 million tons [2]. In the period 1961–2016, the growth in the global fish supply had a notable increase, with an average annual growth rate of 3.2%, higher than the growth rate of world’s population (1.6%), even surpassing that of meat from land animals (2.8%). World fish consumption per capita increased from 9.0 kg in 1961 to 20.2 kg in 2015, and preliminary estimates indicate further growth for 2016 and 2017 to approximately 20.3 kg and 20.5 kg, respectively [3]. Consequently, also the amount of fish waste has undergone a dramatic increase across the world. Currently, the post-catch fish losses represent a huge economic and environmental concern occurring in most fish distribution chains, with large amounts of landed fish lost or discarded between landing and consumption [2]. It is important to underline that today the expansion of consumption is driven not only by the increase in production, but also by different factors, including the reduction of wastage. In fact, although the continuous increase in fish consumption in 2016 a small decrease in global capture fisheries production (90.9 million tons) was observed, compared to the previous two years [3]. In order to promote environmentally, economically, and socially sustainable EU fishing practices, the main recent objectives pursued by the EU Common Fisheries Policy (CFP) are the drastic reduction/prohibition of discards and the use of the captured biomass as best as possible [4].

To date, fish waste is partly destined for the production of fishmeal, fertilizers, and fish oil with low profitability or utilized as raw material for direct feeding in aquaculture [5,6,7,8], and partly thrown away [9].

For this reason, a better fish-waste management is needed to overcome environmental issues and for the fully use of biomass for purposes of high-commercial value, at the same time. In this context, the growing recent attention to alternative uses of fish byproducts plays an important role in the economic growth and sustainable development. Several studies have been reported to analyze their possible uses, as they represent a rich source of value-added compounds, including enzymes, bioactive peptides, and bio-polymers, with many possible uses in several fields [10,11].

In this review, the authors discuss about how circular bioeconomy can be achieved through sustainable fish waste management, examining the global situations of fish waste, the main characteristics of compounds obtained from fish byproducts, and possible uses for production of value-added products.

## 2. Fish Waste in the Circular Bioeconomy Era

The enormous population growth that has taken place in the last two decades, and the consequent extensive use of nonrenewable resources, has negatively affected the quality of the environment and pushed towards sustainable strategies. In this context, the uses of alternative resources that can replace fossil ones and the development of renewable processes based on sustainability are essential for future generations. The transition from a linear to a circular economy is currently an indispensable aspect for managing resources in an eco-efficient way, since the concept of sustainability is totally based on the circularity of all the necessary materials.

The circular bioeconomy is an integral part of the circular economy and it is fundamental in achieving both sustainability in terms of resources and environmental sustainability. The bioeconomy uses materials of biological origin and imitates or uses processes developed by nature to achieve an efficient use in terms of resources [12]. According to the European Commission, bioeconomy is defined as “the production of renewable biological resources and the conversion of these resources and waste streams into value-added products, including food, feed, bio-based products and bioenergy” [13]. In this way, the anthropogenic consumption of raw materials of fossil origin is reduced, the influx of renewable resources is supported, and the environmental impact is reduced to a minimum.

The greatest strengths of the circular bioeconomy are: awareness by people and industry, involvement of stakeholders and policy makers, support of politics, sustainable production and consumption, resources valorization, and zero waste. In this sense, the bio-waste valorization approach plays a fundamental role in bringing circularity to the bioeconomy. In this area huge efforts are underway by the scientific community, together with the support of the government, and are largely directed at the recovery of resources from biological waste.

In this context, a growing awareness and special attention to the development of greener and more sustainable processes have led to a greater interest in the use of unwanted marine resources, such as the huge amount of waste obtained by fishing and aquaculture, which is a very promising source of products with high market value [10,11,14]. Each year, a huge amount of biomass is (i) discarded, generally incinerated, increasing the energy consumption, financial cost, and environmental impact of their management process [15], or (ii) utilized for low-value products; to date, fish waste is used mainly in the fish meal industry, since it contains almost the same amount of proteins as fish meat [5,6,7]. Moreover, the nutritional composition of fish waste allows to supply plant nutrients or to enrich a compost. In fact, fish waste is/can be processed to produce several fertilizers [8], and currently commercial fish-based fertilizers are used for agricultural and horticultural crops [16]. In addition, fish waste has high concentration of biodegradable organics which could be recycled as attractive co-substrate for waste activated sludge, to improve the methane production during anaerobic co-digestion [17,18].

More than 70% of the total fish caught is subjected to further processing before being placed on the market [19], resulting in the production of large amounts (approximately 20–80%) of fish waste, depending on the level of processing (e.g., gutting, scaling, filleting) and species, because each species has a specific composition, size, shape and intrinsic chemistry [20,21]. These operations generate discards which mainly include muscle-trimmings (15–20%), skin and fins (1–3%), bones (9–15%), heads (9–12%), viscera (12–18%), and scales (5%) [22]. Fish processing is an important need for large fish companies both to reduce the costs related to transport of inedible parts of the fish and to increase stability and quality of products, removing parts, such as the viscera, that might contain bacteria and enzymes, which represent a risk for processing and storage of the fish [23]. Preserving the nutritional quality of products represents one of the main challenges for the industry. The degradation of proteins by enzymes is a key aspect that should be minimized, as a high degree of hydrolysis could produce bitter-tasting peptides [23,24], and together with the lipid peroxidation, lead variability in raw materials [24,25]. Acceptable levels of lipid and protein hydrolysis are enormously dependent on the product, based on its end use [26,27]. Hydrolysis determines structural and conformational changes, with possible negative effects on the physicochemical and functional properties of proteins [23]. For this reason, the control of the autolysis and auto-oxidation of these products is fundamental for their use [20]. This represents a real challenge for fishing vessels, which require more advanced equipment and technologies for capture and better handling, necessary to maintain the quality of the byproducts [28,29]. Moreover, to ensure food safety and consumer protection, increasingly stringent hygiene measures have been implemented at the National and International level.

In addition, a substantial amount of by-catch is rejected each year, including both low-value species and tons of commercially valuable but undersized fish. However, the amount of fish byproducts is expected to increase in the next years, due to the implementation of the landing obligation, as part of the new reform of the EU-CFP. It requires the obligation of landing all commercially exploited species under the total allowable catch regulations, including undersized fish which cannot be used for direct human consumption, and endangered species, avoiding the great waste of precious fish biomass through the practice of fish discards [9,30]. In fact, it has been estimated that nearly 25% of all the caught fish never reaches the market; every year, approximately 27 million tons of unwanted fish are discarded into the sea, and much of it does not survive.

The final goals of the new landing obligation are also closely related to two other EU strategies, Blue Growth and the 2020 EU Strategy, involved in developing sustainable socio-economic and environmental growth in the marine and maritime EU region [31]. In particular, Blue Growth is a long-term strategy based on the exploitation of seas and oceans, which have great potential for innovation and growth, offering new ways to help the EU emerge from its current crisis and drive the economy. Considering all the activities that depend on the sea, the EU blue economy implies 5.4 million jobs and a gross added value of almost €500 billion per year. The Organization for Economic Co-operation and Development (OECD) expects that many ocean-based industries have the potential to outperform the global economy by 2030, in terms of value added and jobs, and the output of the global ocean economy could be more than double [32].

Part of the Blue Growth strategy is focused on the blue biotechnology that involves the transformation of raw marine materials in order to obtain products with high economic value useful for different biotechnological applications, which could be utilized for the development of innovative markets, contributing to the goals of the EU strategies.

In this context, the valorization strategies for fish discards and fish byproducts could contribute to the economic growth. Likewise, new uses for fish waste could alleviate the costs associated to the landing obligation, and reduce the enormous environmental problems associated to the large amount of waste.

## 3. Fish Byproducts as Source of High Added-Value Compounds

Due to this massive increase in fish consumption, in 2015 fish accounted for approximately 17% of the global population’s intake of animal protein and 6.7% of all proteins consumed in the world [3]. A fish-based diet has a huge positive nutritional impact, also playing an important role in correcting unbalanced diets and countering obesity. In fact, besides being a rich source of high-quality protein carrying all essential amino acids, fish supply (i) essential fats, such as long-chain omega-3 fatty acids, (ii) vitamins, such as D, A and B, and (iii) minerals, including calcium, iodine, zinc, iron and selenium. Thanks to these valuable nutritional properties, fish bring health benefits in protecting against cardiovascular disease, and help fetal and infant development of the brain and nervous system [2]. In fact, marine species, including fish products, are popular for medical use and are considered as an exploitable source of efficacious animal-derived medicinal products (ADMPs) [33].

Fish byproducts are a nutritionally important source of proteins, fatty acids, and minerals, as their composition is similar to that of fish fillet and other food products used for consumption. Studies conducted on meagre and gilthead sea bream fish species have shown that skin is the most significant protein source, trimmings and bones are rich in calcium, and the head, intestines, and bones are a good source of lipids [34]. Specifically, the mean value of all fish byproducts, calculated on a dry weight basis, is 49.22–57.92% for protein content, 21.79–30.16% for ash content, and 7.16–19.10% for fat content [35,36,37]. The latter value is slightly higher than those found by [34].

In addition to the share of world fish production utilized for direct human consumption, the use of byproducts is increasingly gaining attention, as they offer a significant and sustainable source of high-value bio-compounds, due to their high content of collagen, peptides, chitin, polyunsaturated fatty acids (PUFAs), enzymes, and minerals, suitable for biotechnological or pharmaceutical applications with high market value [10,11,14,38,39,40,41,42,43,44].

Many studies have already extensively reported the methods used for the extraction of these compounds [14,38,45,46,47]. For this reason, this paragraph provides an overview of the extraordinary potential of fish byproducts, listing the possible applications found so far for the most interesting compounds extracted from fish byproducts (e.g., collagen, peptides, chitin, oil, enzymes).

### 3.1. Collagen 

The Marine Collagen Market has been estimated to reach USD 983.84 million by 2025, growing at a Compound Annual Growth Rate (CAGR) of 7.4%. The growth of the marine collagen market is due to the use of collagen in the cosmetic, food and beverage industry [48], and fish waste represents a huge and cheap source of collagen for the industry [49].

Collagen is a complex and abundant structural protein, that is present only in metazoa and groups approximately 20–30% of animal proteins [50,51]; 28 different types of collagen have been found in higher metazoan [52]. Collagen types are classified depending on their primary structures and forms of supramolecular organization. Fibrillar collagens (collagen types I, II, III, V, XI, XXIV, XXVII) participate in the formation of cross-striated fibrils. In mature fibrillar collagen molecules, the collagenous domain of each α chain contains the repeated triplets sequence Gly-X-Y. Fibrillar collagen (type I) is the most abundant protein in vertebrates and it plays a key role in the mechanical properties of bones, tendons, and skin [53]. Other subfamilies correspond to non-fibrillar collagens, which contain different interruptions in their collagenous domains. Non-fibrillar collagens may be classified in several subfamilies: (1) network-forming collagen which includes basement membrane (type IV), beaded filaments forming collagen (type VI), anchoring fibril (type VII), short-chain collagens (types VIII, X), which are hexagonal network-forming collagens; (2) fibril-associated collagens with interrupted triple helices (types IX, XII, XIV, XX); (3) fibril-associated with interrupted triplex helix-like (types XVI, XIX, XXI, XXII); (4) plasma membrane collagens (types XIII, XVII, XXIII, XXV); (5) multiplexin collagens (types XV, XVIII), and other molecules with collagenous domains (types XXVI, XXVIII) [14,53].

Partial hydrolysis and heat denaturation of collagen lead to gelatin production. Generally two types of gelatin, type A and B, are obtained by acid and alkaline hydrolysis, respectively [54]. During production of gelatin by hydrolysis the inter-molecular and intra-molecular bonds of collagen fibers are broken [55], and the structural characteristics of gelatin depend on the distribution of molecular weight (MW), structure and composition of each subunit [54]. Composition of gelatin is closely related to the origin collagen, however it contains many glycine, proline and hydroxyproline residues, approximately 21%, 12%, and 12%, respectively [56].

Moreover, hydrolysed collagen can be obtained by enzymatic action (alcalase, papain, pepsin, and others) in alkaline or acid media at specific incubation temperature, generally above 40 °C. The obtained peptides have low MW (0.3–8 KDa) and are easy to be absorbed by the intestine and available for tissues [57]. Their solubility and biological activity are related to the parameters used for the extraction, such as types of used enzymes and type of hydrolyses [58]. Recently, it has been shown that green and sustainable deep eutectic solvents can also be used to extract collagen peptides [59].

The main sources of commercial collagen are bovine and pig skin, cattle bones and other mammalian animal waste [60]. Collagen is also present in marine animals like sponge, jellyfish, molluscs (mussel, squid cuttlefish and octopus), echinoderms, and fish [61]. Therefore, fish waste could be used to obtain cheapest collagen. In fact, it mainly consists of bones, skin, scales and fins [5] having high collagen content, thus it may be considered to produce collagen [14]. Fish collagen has important properties, as such as a more efficient adsorption into the body (up to 1.5 fold) and higher bioavailability with respect to the porcine and bovine collagen [62]. However, fish collagen possesses disadvantages in melting point, mechanical strength, biomechanical stiffness, and high rate of degradation [63]. However, the main problem related to the use of fish collagen is the development of a sustainable extraction process for its commercial exploitation [64].

Different procedures have been proposed to extract collagen from fish waste. Acid extraction procedure is the most common method to extract collagen from fish waste, and the obtained product is referred to as acid-soluble collagen (ASC). Collagen extraction is obtained using acids, such as acetic acid or HCl, which solubilized the collagen chains, improving the extraction yield. Pepsin extraction procedure consists in the treatment of the collagen fibres with the pepsin to cleave the specific regions in order to promote solubilisation and increase collagen extraction yield; in this case, the extracted collagen is named pepsin-soluble collagen (PSC) [49].

Collagen and gelatin properties are of huge importance for many application fields, in fact recent studies are pointing towards the use of food sources such as fish-derived collagen and gelatin as excellent functional molecules for the cosmetic, pharmaceutical, biomaterial, food, and nutraceutical industries [14,58,65]. Thus, in the next section and in Table 1 we discuss the recent advances in the use of collagen and gelatin from fish byproducts.

#### Fish Collagen Applications

Collagen is the most promising natural biomaterial used to build scaffolds in tissue engineering. In fact, scaffolds play a key role in promoting cell regeneration, as they are involved in cell adhesion and differentiation processes [66]. Synthetic polymeric scaffolds have been widely used in cartilage regeneration [67]. However, several limitations, such as low biocompatibility and high rate of immune reactions, reduce their further application [68]. On the contrary, natural scaffolds, including collagen and gelatin, from a wide range of marine sources may possess several advantages, such as biosafety, higher biocompatibility, and weak antigenicity. Moreover, natural scaffolds contain amino acid residues, that promote cell adhesion and differentiation [69]. Tilapia skin collagen has been tested as scaffold in cartilage regeneration [70]. Collagen was found to promote cartilage production in vitro (chondrocytes from rabbit auricula) and in vivo (rabbit). In addition, in situ cartilage repair in rabbit articular defect model collagen was tested. Collagen, also in this case, was found to promote cartilage regeneration. Cartilage repair in situ is a direct evidence used to predict the potential of future clinical applications. Tilapia skin collagen did not exhibit cytotoxic side effects in vitro and inflammatory response in mice model. Similar results were obtained using electrospun nanofibrous membranes of fish collagen/polycaprolactone (FC/PCL) [66]. Collagen from tilapia scale was used to obtain electrospun membranes. The obtained membranes were able to induce in vitro formation of preliminary cartilage-like tissue displaying typical lacunae structures. FC/PCL membranes have been shown to promote in vivo cartilage regeneration in rabbit model. Oh et al. [71] also proposed a FC/PCL composite scaffolds for bone regeneration processes. Collagen obtained from *Paralichthys olivaceus* skin was used to produce FC/PCL scaffolds, able to induce cell differentiation, calcium deposition and mineralization in vitro, in model of mouse mesenchymal stem cells. In addition, in vivo experiments were performed to implant the scaffolds in a rabbit tibia. FC/PLC scaffolds have been found to improve new bone formation and no inflammatory or immune responses were observed after scaffolds implantation [71]. Suzuki et al. engineered oral mucosa by using collagen extracted from tilapia scale. Primary oral mucosa keratinocytes were grown on collagen scaffold. The authors also reported a fully differentiated and stratified epithelial layer developed on collagen scaffolds [71]. Moreover, blended hydrogels composed of polyvinyl alcohol (PVA) and fish collagen (FC) were used as scaffold to human periodontal ligament fibroblasts (HPDLFs) and gingival fibroblasts (HGFs) growth. A mixture of PVA/FC 50:50 increased growth of both HPDLFs and HGFs [72]. Collagen from Nile tilapia scales also promoted matrix mineralization in in vitro models [73]. These data suggest that the tilapia collagen scaffolds may be applied in the production of tissue-engineered oral mucosa equivalents for clinical use. Mineralized collagen from the skin of Atlantic salmon (*Salmo salar*) has been used to produce scaffold for bone regeneration. It allowed human mesenchymal stem cells to grow and to be responsive to the osteogenic stimuli [74]. Moreover, collagen from raw cartilage from shark (*Prionace glauca*) and ray (*Zeachara chilensis* and *Bathyraja brachyurops*) has been shown to possess promising properties and biotechnological potential in order to regenerate damaged cartilaginous tissues [75]. Cao et al. [76] produced a scaffold-controlled release system for tissue skin engineering based on collagen/chitosan.

Another interesting aspect is the use of the collagen as biomaterials in wet wound dressing [77]. Collagen from *Oreochromis niloticus* skin was used to prepare biomedical hydrogel. It has been proven that collagen hydrogel was able to heal of deep second-degree burn of rat skin [78]. Collagen accelerates wound healing by promoting the expression of vascular endothelial growth factor (VEGF), transforming growth factor-beta (TGF-ß1), basic fibroblast growth factor (bFGF), and alpha-smooth muscle actin (α-SMA) [79]. Scale fish collagen was also used to deliver polymyxin B, which displays antibacterial activity against Gram-negative bacteria, such as the multidrug-resistant *Pseudomonas aeruginosa* [80] and bacitracin, that displays antibacterial activity on Gram-positive bacteria, such as multidrug-resistant staphylococci [81], in order to prevent bacterial infection during wound healing processes [82]. Moreover, collagen-based scaffolds impregnated with sago starch capped silver nanoparticles (AgNPs) were fabricated by using collagen derived from fish scales of *Lates calcarifer*. In vitro studies indicated high tensile strength values for their use as wound dressing materials and antibacterial activities against both Gram-positive (*Staphylococcus aureus*) and negative (*Escherichia coli*) bacterial strains [83]. Moreover, collagen/bioactive glass nanofibers prepared with collagen from tilapia skin, promoted wound healing and skin regeneration, and at the same time reduced bacterial infection in rat model, showing antibacterial activity against *S. aureus* [84]. In addition, Ibrahim et al. [85] tested Nile tilapia (*O. niloticus*) skin in wound healing in a donkey model. Fish skin accelerated the wound healing process and efficiently inhibited the local microbial activity. Moreover, sponge scaffolds developed by using porcine skin-derived collagen and *Ctenopharyngodon idellus* scale-derived collagen were compared. Both scaffolds efficiently promoted skin regeneration in rabbit model, with good wound-healing outcome compared to gauze and Vaseline gauze groups. Results showed that fish-collagen scaffold could be an alternative candidate to bovine-collagen scaffolds in burn wound care applications [86].

Several studies pointed out the antioxidant activity of collagen from fish, interesting for several applications in different fields, including food preservation, healthcare and cosmetics. The antioxidant capacity was mainly evaluated by using the assays 2,2-diphenyl-1-picrylhydrazyl (DPPH), 2,2′-azinobis-(3-ethylbenzothiazoline-6-sulfonate) (ABTS), oxygen radical absorbance capacity (ORAC), Ferric-reducing antioxidant power (FRAP).

PSC obtained from the skin of *Lophius litulon* was also tested. It has been shown to possess both antioxidant properties in vitro based on its ability to scavenge different free radicals, such as DPPH, ABTS·, HO·, and O_2_- and increase the content of superoxide dismutase (SOD) and catalase (CAT), and wound healing activity in mice model [87]. Collagen from red tilapia *Oreochromis* sp. skin was also tested for its antioxidant properties [88]. It has been found that ASC, salt soluble collagen, and pepsin-hydrolysed collagen showed both radical scavenger activity and ferric-reducing antioxidant power. Hua and collaborators [89] prepared a bioactive collagen/chitosan complex using fish skin collagen and chitosan solution as raw materials. The bioactive complex possessed antioxidant activity when tested in vivo in mice model, it reduced malonic dialdehyde content and increased SOD activity in mice serum [89]. Moreover, collagen/gelatin/chitosan novel porous scaffolds fabricated using blends of collagen and gelatin obtained from the marine big eye snapper *Priacanthus hamrur* skin showed both antioxidant (maximum activity in both DPPH and ABTS assay at 1mg/mL) and antibacterial (maximum activity at 200 µg by disc diffusion assay) activity against *E. coli* and *S. aureus* [90].

In addition, *Nibea japonica* swim bladders [91], yellowfin tuna (*Thunnus albacares*) skin [92], *Scomber japonicus* bone and skin [93], shark cartilage [94] also showed antioxidant properties.

Collagen from fish waste could also find application in the food industry as a food additive and packaging [14]. Fish collagen has been used as yogurt additive [95], collagen affected proteolysis of milk proteins and conferred angiotensin I-converting enzyme (ACE) inhibitory property; ACE has a key role in the regulation of blood pressure in mammals and in development of cardiovascular disease, therefore it represents an important target in the treatment of high blood pressure [96]. 

Moreover, PSC extracted from *P. glauca* skin was used to produce chitosan–collagen composite coating. The composite coating has been found to preserve important properties of *Pagrus major* fillet quality, such as drip loss, pH, and to reduce microbial growth during storage at 4 °C [97]. Collagen skin from *Mustelus mustelus* combined with chitosan has been used to produce a protective film in order to preserve nutraceutical products. In addition, this film had antioxidant activities and could act as anti-UV barriers [98].

Many other fish byproducts collagen applications have been found in the hydrolyzates/peptides obtained from the digestion of collagen and gelatin, which will be discussed in the next section (Section 3.2).


marinedrugs-19-00116-t001_Table 1Table 1Collagen/gelatin from fish waste and their possible applications. BW, body weight; PVA, polyvinyl alcohol; FC/PCL, fish collagen/polycaprolactone.CompoundByproductSourceApplicationsActivity/Concentration UsedReferenceCollagen SkinTilapiaCartilage tissue engineering2% weight/volume[70]Collagen/polycaprolactoneScale *Oreochromis* sp.Cartilage tissue engineering12% weight/volume[66]Collagen/polycaprolactoneSkin
*Paralichthys olivaceus*
Bone regenerationNot specified[71]CollagenScale*Oreochromis* sp.Tissue-engineeredoralmucosa1% weight/volume[71,99]Collagen/polyvinyl alcoholNot specified*Oreochromis* sp.Human periodontal ligament fibroblasts (HPDLFs), gingival fibroblasts (HGFs)Not specified[72]CollagenScales
*Oreochromis niloticus*
Bone regeneration0.3%, weight/volume[73]CollagenSkin
*Salmo salar*
Bone tissue engineering0.2–2.5 mg/mL[74]CollagenCartilage
*Prionace glauca*

*Zeachara chilensis*

*Bathyraja brachyurops*
BioscaffoldNot specified[75]Collagen/chitosanSkin
*Hypophthalmichthys molitrix*
Skin regenerationNot specified[76]CollagenSkin
*Oreochromis niloticus*
Wound healing15 mg/mL[78]CollagenScale
*Larimichthys crocea*
Wound healing5–15% weight/volume[82]Collagen nanofibersSkin*Oreochromis* sp.Skin regeneration, antibacterial 30 mg/kg BW[84]CollagenSkin *Oreochromis* sp.Wound healingNot specified[85]CollagenScale
*Ctenopharyngodon idellus*
Wound healingNot specified[86]CollagenSkin
*Lophius litulon*
Antioxidant, wound healing 1–8 mg/mL4 g/day[87]CollagenSkin*Oreochromis* sp.Antioxidant 2–7 mg/mL[88]Collagen/chitosanSkinNot specifiedAntioxidant 12–100 μg/mL[89]Collagen/gelatinSkin
*Priacanthus hamrur*
Antioxidant, antibacterialDPPH and ABTS 1 mg/mLAntibacterial 200 μg[90]CollagenSwim bladders
*Nibea japonica*
Antioxidant12.5–50μg/mL[91]Collagen/gelatinCartilage
*Carcharhinus albimarginatus*
Antioxidant1–5 mg/mL[94]CollagenSkin
*Scomber japonicus*
Antioxidant2.5–10 mg/mL[93]Collagen/gelatinScale
*Katsuwonus pelamis*
Antioxidant0.1–5 mg/mL[100]CollagenSkin
*Prionace glauca*
Food packaging2.5% weight/weight[97]CollagenSkin
*Mustelus mustelus*
Food packaging0.1% weight/volume[98]


### 3.2. Peptides

Fish protein hydrolysates market size was about USD 420 million globally in 2019 and it is supposed to increase of 4.5% CAGR between 2020 and 2026 [101]. Several studies have shown that marine organisms may be an excellent source of bioactive proteins/peptides [102]. A good amount of proteins has been isolated from marine processing waste. Hydrolysates and purified peptides have been isolated from several fish species (e.g., pollack, sole, salmon, skate, halibut, tuna, catfish, ray, crocker, turbot and hoki), from whole body fish waste or from specific body part waste, such as frame, scale, bone, head, gonads and viscera (as reviewed by [38,103]). Before they can be effectively used, several extraction methods have been applied in order to get bioactive peptides, such as acid-alkaline hydrolysis (by using acidic or alkaline reagent), enzymatic hydrolysis (by using enzymes to hydrolyze fish waste), and fermentation (by using microorganisms as source of enzymes) [38,47,104]. Enzymatic hydrolysis is especially preferred in food and pharmaceutical industries because the process does not leave residual organic solvents or toxic chemicals [105] and is considered the preferred method to hydrolyze fish skin without losing nutritional value [47]. Fermentation is considered a more natural procedure for protein hydrolysis, and it has been used for centuries as a traditional preservation method, enhances flavor and taste of food but also increases its nutraceutical value [39]. After the extraction process, various purification steps could follow. For example, ultrafiltration (UF), nanofiltration (NF), and gel filtration (GF) are used to purify peptides based on their MW [106], ion exchange chromatography (IEC) is used to purify peptides based on their net charge and reversed-phase HPLC can be used to separate compounds based on their hydrophobicity and hydrophilicity. Peptide sequences are then characterized by mass spectrometry methods such as matrix-assisted laser deionization time-of-flight (MALDI-TOF), electrospray ionization mass (ESI), matrix-assisted laser desorption/ionization mass spectrometry (MALDI-MS) (as reviewed by [38]). For example, Vázquez and collaborators [107] have just published a study on the use of the turbot Scophthalmus maximus byproducts. In particular, turbot byproducts (from turbot head, viscera, trimmings and frames) were subjected to alcalase hydrolysis obtaining fish protein with high yield of digestion, remarkable degrees of hydrolysis and high content of soluble protein. In the paragraph below and in Table 2, we summarize peptides isolated from fish waste and their bioactivities.

#### Fish Peptides Applications

Several peptides derived from fish hydrolysates have been sequenced and available sequences are summarized in Table 2. In addition, hydrolysates and purified peptides have been tested for bioactivities useful for prevention and treatment of several human pathologies. Various authors have reported that amino acid type, position and hydrophobicity have been considered to play a relevant role in peptide bioactivities [38]. Common bioactivities are antimicrobial, antihypertensive, antioxidant and neuroprotective activities [104].

Regarding antimicrobial activities, peptides from fish hydrolysates were mainly active against Gram-negative bacteria, such as *Aeromonas hydrophila*, *Klebsiella pneumonia*, *Salmonella enterica*, and *Salmonella typhi,* and Gram-positive bacteria, as *Streptococcus iniae*, *Micrococcus luteus*, *S. aureus*, and *Bacillus cereus.* However, antifungal activities have been reported as well, such as against the fungus *Candida albicans*. Activities were evaluated and measured by reporting the minimum inhibitory concentration (MIC) or the minimum effective concentration (MEC) values. SJGAP isolated from Skipjack tuna (*K. pelamis*) skin had the highest activity against *E. coli* (MEC 2.7 μg/mL) and *B. subtilis* (MEC 3 μg/mL) [108], YFGAP from yellowfin tuna (*T. albacares*) was most active against *B. subtilis* (MEC 1.2 μg/mL), *E. coli* (MEC 3 μg/mL), and *V. parahaemolytics* (MEC 3.2 μg/mL) [109]. GKLNLFLSRLEILKLFVGA from Yellow catfish (*Pelteobagrus fulvidraco*) skin and GWGSFFKKAAHVGKHVGKAALTHYL from winter flounder *Pleuronectes americanus* skin also were most active against *B. subtilis* (MIC 2 μg/mL and 1.1–2.2 μM, respectively) [110,111].

Moreover, fish collagen-derived peptides, SIFIQRFTT, RKSGDPLGR, AKPGDGAGSGPR and GLPGLGPAGPK, isolated from *Scomber scombrus* showed antibacterial activity. In particular, GLPGLGPAGPK had antibacterial activity against both Gram-positive and Gram-negative bacteria [112]. In addition, the peptide KVEIVAINDPFIDL from *Scomber scombrus* exhibited antibacterial activity against *Lactobacillus acidophilus*, *Listeria ivanovii*, *Listeria monocytogenes, M. luteus* and *Bacteroides thetaiotaomicron* (MIC 0.263, 0.131, 0.131, 0.26 and, 0.263 mM, respectively) [113].

Protein hydrolysate prepared from tilapia waste showed also resistance against *Vibrio anguillarum* in silver pompano (*Trachinotus blochii*), as well as improved growth performance, metabolism, and innate immune response. The reduced antioxidant enzyme (SOD) activity indicated the fish hydrolysate scavenging activity, which minimizes the necessity of expression of SOD in different tissues [114].

Several fish collagen-derived peptides showed anti-hypertensive activity, by mainly acting with ACE inhibitory activity or having antihypertensive effects on spontaneously hypertensive rats; these peptides are listed below: GPL and GPM (IC_50_ = 2.6 and 17.1 μM, respectively) [115] and PGASTRGA (IC_50_ = 14.7 μM) [116] isolated from *T. chalcogramma* skin, DPALATEPDPMPF purified from *O. niloticus* (higher activity at 20 μg/mL) [117], GPEGPAGAR and GETGPAGPAGAAGPAGPR from *O. niloticus* skin gelatin (inhibition at 5 mg/mL) [118], MVGSAPGVL and LGPLGHQ from Skate (IC_50_ 3.09 and 4.22 μM, respectively) [119], GASSGMPG and LAYA purified from *Gadus macrocephalus* skin gelatin hydrolysates (IC_50_ = 6.9 and 14.5 μM) [120]. Moreover, tripeptides based on GPL isolated from skin gelatin hydrolysates of Alaska Pollock *T. chalcogramma* (containing glycine, proline, and leucine), were synthesized and showed high ACE inhibitory activity [121]. Collagen peptides extracted from *T. chalcogramma* skin, which were obtained by simulating gastrointestinal digestion possessed antihypertensive properties (IC_50_ = 0.49 mg/mL) [122]. In addition, the synthetic peptides, GIPGAP and APGAP, derived from collagen proteolysis of *Raja clavata* skin have been also characterized for their ACE-inhibitory properties (IC_50_ = 27.9 and 170.2 μM) [123].

Fahmi et al. [124] tested hydrolysates by using spontaneously hypertensive rats fed with 300 mg of peptides (kg of body weight)^−1^ d^−1^ and also purified and tested ACE inhibitory activities of four peptides (GY, VY, GF, VIY) from sea bream scales (IC_50_ 265, 16, 708, 7.5 µM, respectively). Peptides were also isolated from *Raja kenojei* skin protein hydrolysates obtained by treatment with several proteases (i.e., alcalase, a-chymotrypsin, neutrase, pepsin, papain, and trypsin). Peptides obtained by chymotrypsin digestion, PGPLGLTGP and QLGFLGPR, showed the higher ACE-inhibitory activity(IC_50_ = 95 and 148 μM, respectively) [125]. Moreover, GLPLNLP was isolated from salmon *O. keta* skin trypsin hydrolysate (IC_50_ = 18.7 μM). Synthetic peptides based on the sequence of the purified peptide from salmon skin were also tested, and the peptide GLP had the highest ACE inhibitory activity (IC_50_ = 9.08 μM) [126]. Interesting peptides with ACE inhibitory activity were isolated from fish frame protein hydrolysates, including FGASTRGA isolated from Alaska pollack protein hydrolysates, obtained with pepsin (IC_50_ = 14.7 μM) [117], MIFPGAGGPEL obtained from Yellowfin sole (*Limanda aspera*) hydrolysates MW < 5KDa (IC_50_ = 28.7 μg/mL) [127], GDLGKTTTVSNWSPPKYKDTP isolated from tuna (IC_50_ = 11.28 μM) [128].

Ohba et al. [129] evaluated the ACE inhibitory and antioxidant activities of yellowtail fish bone hydrolysates. In particular, they studied the activity of crude hydrolysates, hydrolysates with MW ≥ 10,000, MW between 1000 and 10,000, or MW ≤ 1000. They found that IC_50_ were 2.0, 2.8, 1.9 and 1.5 mg/mL for crude hydrolysates, hydrolysates with MW ≥ 10,000, MW between 1000 and 10,000 and MW ≤ 1000, respectively, showing that hydrolysates with MW ≤ 1000 were the most active. Regarding the antioxidant activity, the most active were hydrolysates with MW ≥ 10,000, followed by crude hydrolysates, MW between 1000 and 10,000, and MW ≤ 1000 (IC_50_ were about 5, 8, 10, 35 mg/mL). Similar trend was observed for yellowtail fish scale hydrolysates [129]. Similarly, Jeon et al. [130] analyzed ACE inhibitory and antioxidant activity of cod frame protein hydrolysates of 30, 10, 5 and 3 KDa. The 3 KDa hydrolysates had the highest ACE inhibitory activity (IC_50_ = 0.08 mg prot/mL), while the 5 KDa hydrolysates had the highest antioxidant capacity (~40% oxidant inhibitory ratio).

As reported by Theodore and colleagues [131], low MW peptides have generally higher ORAC values while high MW peptides have higher FRAP and DPPH radical scavenging activities [131]. However, it is sometimes difficult to compare bioactivities between different peptides analysed with different techniques because radical scavenging activities are not always reported with the same units. Recently, Vázquez et al. [107], in order to valorise farmed fish processing wastes, studied fish protein hydrolysates obtained by alcalase hydrolysis from turbot *S. maximus* head, viscera, trimming and frames. Antioxidant and antihypertensive possible activities were evaluated and results showed that viscera hydrolysates, containing peptides above 1000 Da and below 200 Da, were the most active with scavenging activity of 65.15% (DPPH), 12.81 µg BHT/mL (ABTS), 8.03 µg Trolox/mL (Crocin) and 81.9% of ACE inhibitory activity. 

Other peptides from fish skin showed antioxidant properties, such as PYSFK, GFGPEL and VGGRP from the grass carp *Ctenopharyngodon idella* [132], AVGAT from Thornback ray [123], DPALATEPDMPF, EGL and YGDEY from Nile tilapia *O. niloticus* [133,134], PFGPD, PYGAKG, YGPM from Spanish mackerel *Scomberomorous niphonius* [134], GSGGL, GPGGFI and FIGP from blue leatherjacket *Navodon septentrionalis* [135], GATGPQGPLGPR, VLGPF and QLGLGPV from seabass *Lates calcarifer* [136], FDSGPAGVL and DGPLQAGQPGER from Jumbo squid *Dosidicus gigas* [137], PAGT from Amur sturgeon [138], P1 and P2 from pollack [139], and LSGYGP from the tilapia *O. niloticus* [140]. Finally, generic hydrolysates from snappers *Priacanthus macracanthus* and *Lutjanus vitta* and sole skin [141,142,143] and hydrolysates from the herring *Clupea harengus* whole, body, head and gonads had antioxidant activity [144]. TCSP and TGGGNV from the cod *Gadus microcephalus* showed both anti-hypertensive and antioxidant properties [145].

Fish frame protein hydrolysates are also a rich source of antioxidant peptides, including GSTVPERTHPACPDFN isolated from Hoki (*Johnius belengerii*) [146], N-terminal RPDFDLEPPY purified from yellowfin sole (*L. aspera*) [147], LPHSGY isolated from Alaska pollack hydrolysates MW < 1KDa [148]. Moreover, antioxidant activity was identified in the peptide VKAGFAWTANQQLS isolated from tuna backbone protein, that inhibited the lipid peroxidation in linoleic acid emulsion system and quenched free radicals (DPPH, hydroxyl and superoxide) in a dose-dependent manner [149].

Several studies pointed out the antioxidant activity of hydrolysates of collagen and its derivatives, obtained from different fish byproducts, including *Gadous macrocephaius* skin [150], shark cartilage [94], yellowfin tuna (*T. albacares*) skin; particularly low-weight peptide (< 3kDa) showed higher radical scavenging properties [92]. Collagen polypeptide from tilapia skin (MW< 3000Da) had protective effects against injuries to the liver and kidneys of mice induced by d-galactose by reducing oxidative stress [151]. Moreover, collagen-derived peptides that have shown antioxidant activity were: GLFGPR peptide from *L. calcarifer* skins [152], HGPLGPL peptide from *J. belengerii* skin [137], GPRGTIGLVG, GPAGPAG, and GFPSG from scales of *Pseudosciaena crocea* [153], HGPHGE, DGPKGH, and MLGPFGPS from *Katsuwonus pelamis* scales [100], GPDGR, GADIVA, GAPGPQMV, AGPK, and GAEGFIF from *K. pelamis* bones with GADIVA and GAEGFIF that showed strongest antioxidant activity [154], GPE, GARGPQ, and GFTGPPGNG from *Sphyrna lewini* cartilage [155], YGCC, DSSCSG, NNAEYYK, and PAGNVR purified from *Theragra chalcogramma* skin [156]. Moreover, peptides NHRYDR and GNRGFACRHA from *Magalapis cordyla* skin and *Otolithes ruber* skin, respectively, reduced PUFAs peroxidation [157].

Regarding neuroprotective activities, QGYRPLRGPEFL isolated from Skate (*R. kenojei*) skin showed β-Secretase inhibitory activity (IC_50_ value of 24.26 μM, [158]), while collagen hydrolysates isolated from Salmon (*O. keta*) skin had anti-acetylcholinesterase activity, learning and memory effects [159,160]. Xu et al. [159] investigated the neuroprotective effects of salmon collagen peptides (intragastrically administered at 0.33 g/kg, 1.0 g/kg and 3.0 g/kg body weight) in male rats with perinatal asphyxia. Results showed facilitated early body weight gain, long-term learning and memory, reduced oxidative damage and acetylcholinesterase activity in the brain, and increased hippocampus phosphorylated cAMP-response element binding protein and brain derived neurotrophic factor expression. Salmon learning and memory effects were evaluated also by Pei et al. [160] in 20-month-old female C57BL/6J mice fed with *O. keta* collagen peptides (0.22%, 0.44% or 1.32% wt/wt) compared to aged control mice. Memory effects were evaluated by step-down test, Morris water maze, oxidative stress, expression of brain-derived neurotrophic factor, and postsynaptic density protein 95. The positive results suggested the potential use of the salmon collagen peptide for functional foods to relieve aging memory deficits [160].

Collagen peptides from fish waste are an emerging substance also in the cosmetic field. A randomized triple-blind, placebo-controlled, parallel study was performed to evaluate the efficacy of collagen from *Pangasius hypophthalmus* on skin wrinkles and elasticity. Participants were females between the ages of 45–60 and displayed visible signs of natural and photoaging on their face and they consumed 10 g of hydrolysed collagen powder daily. At week 12, participants in the collagen consuming group reported improvements in overall skin score (9%) and wrinkle (15%), elasticity (23%), hydration (14%), radiance (22%), and firmness (25%) [161]. Moreover, a fish collagen drink was tested for improvement of skin aging. Collagen drink improved cell viability of UVA irradiated human skin fibroblast (CCD-966Sk), their mitochondrial activity was restored and reactive oxygen species (ROS) production triggered by UVA treatment was reduced. Moreover, collagen drink promoted the expression of SOD1, SOD2, CAT genes, and improved both collagen and elastin production [162]. In addition, skin-derived peptides from the sutchi catfish *P. hypophthalmus* showed matrix metalloproteinase (MMP) inhibitory activity [163,164]. In particular, Lu et al. [162] found that cod skin gelatin hydrolysates were especially active at 0.1 mg/mL inhibiting MMP-1 expression in fibroblasts irradiated with UVB of 20 mJ/cm^2^. Two peptides, GEIGPSGGRGKPGKDGDAGPK and GFSGLDGAKGD, were isolated from these hydrolysates and found to exhibit a significant inhibition of MMP-1, phosphorylated extracellular signal-regulated kinase (p-ERK) and p-p38. GEIGPSGGRGKPGKDGDAGPK also inhibited phospho–c-Jun *N*-terminal kinase (p-JNK) in mitogen-activated protein kinase (MAPK) signaling pathways. Pyun et al. [164] analysed the anti-photoaging effects of collagen tripeptide (167–333 mg/kg/day oral administration for 14 weeks) on a hairless mouse model. Mice showed significantly reduced wrinkle formation, skin thickening, transepidermal water loss, skin hydration and prevented UVB-induced MMP-3 and -13 expression, and MMP-2 and -9.

Hydrolysed fish collagen has been found to induce osteoclastin differentiation of bone marrow mesenchymal stromal cells (BMSCs), improving expression levels of anti-inflammatory mediators, such as inteleukin-6 (IL-6), TGF-β1, and prostaglandin E2 (PGE2). Moreover, collagen hydrolysed reduces the expression level of pro-inflammatory mediator, such as interleukin 1 beta (IL-1β) and tumor necrosis factor alpha (TNF-α) [165].

Subcritical water-hydrolysed fish collagen peptide (SWFCP) from tuna skin shown inhibitory activity of adipocyte differentiation, which was followed by decreased expression of CCAAT-enhancer-binding protein-α (C/EBPα), peroxisome proliferator-activated receptor-γ (PPAR-γ), and adipocyte protein 2 (aP2) genes, regulators of differentiation of adipocytes [166]. 

Peptides isolated from only two species, SPGSSGPQGFTG, GPVGPAGNPGANGLN, PPGPTGPRGQPGNIGF from the halibut *Hippoglossus stenolepis* and IPGDPGPPGPPGP, LPGERGRPGAPGP and GPKGDRGLPGPPGRDGM from the tilapia *O. niloticus,* have been reported to have anti-hyperglycemic activity [135]. Daily administration of tilapia skin gelatin hydrolysate (750 mg/kg/day) for 30 days was more potent to improve the glucose tolerance in streptozotocin-induced diabetic rats than hydrolysates from halibut skin (750 mg/kg/day), inducing the inhibition of plasma dipeptidyl peptidase IV (DPP-IV) activity, enhancement of glucagon-like peptide-1 (GLP-1) and insulin secretion. Hydrolysates from frame of the sole *Limanda aspera* were the only one that were reported to have anticoagulant activity [167]. The identified protein of 12.01 KDa, named yellowfin sole anticoagulant protein (YAP), inhibited the activated coagulation factor XII (FXIIa) and antagonized the platelet membrane glycoprotein integrin to arrest platelet aggregation by inhibiting thrombosis *in vitro*. In addition, hydrolysed collagen promoted osteogenic differentiation of bone marrow mesenchymal stromal cells (BMSCs) which preserved their immunomodulation ability. An anticancer peptide, FIMGLY, was also isolated from *Raja porosa* cartilage, and it was found to be active against HeLa cells [168]. Collagen hydrolysates obtained from skin of *Aluterus monoceros* have been found to possess anticancer, antidiabetic and wound healing properties, thus suggest that collagen may represent a flexible and multipurpose material [169].

Oral administration of collagen peptides from *Oncorhynchus keta* skin in growing rats promoted size, mineral density of long bone in treated rats [170]. In addition, collagen peptides from skin of Gadiformes fish species had chondroprotective effects in the osteoarthritis-induced model [171]. Finally, peptides from frame and bone of the hoki *J. belengerii* (VLSGGTTMYASLYAE and a phosphopeptide, named FBP, respectively) and frame of the pollack *T. chalcogramma* (VLSGGTTMAMYTLV) had calcium-binding properties, and peptides (GPAGPHGPPGKDGR, AGPHGPPGKDGR, AGPAGPAGAR) from Pacific cod skin had iron-chelating activity [172,173,174,175].


marinedrugs-19-00116-t002_Table 2Table 2Hydrolysates and purified peptides from fish waste and their possible applications. MW, molecular weight; MIC, minimum inhibitory concentration; MEC, minimum effective concentration; BW, body weight.CompoundByproductSourceApplicationsActivity/Concentration UsedReferenceSJGAPSkinSkipjack tuna (*Katsuwonus pelamis*)AntimicrobialMECs 3, 26, 4.8, 25, 2.7, 9, 16 μg/mL against *B. subtilis, M. luteus, S. iniae, A. hydrophila, E. coli, V. parahaemolytics, C. albicans*[108]YFGAPSkinYellowfin tuna (*Thunnus albacares*)AntimicrobialMECs 1.2, 6.5, 17, 8, 3, 3.2 μg/mL against *B. subtilis, M. luteus, S. iniae, A. hydrophila, E.coli, V. parahaemolytics*[109]GKLNLFLSRLEILKLFVGASkinYellow catfish (*Pelteobagrus fulvidraco*)AntimicrobialMIC 2, 4, 16, 64 μg/mL against *B. subtilis, S. aureus, E.coli, C. albicans*
[110]SIFIQRFTT, RKSGDPLGR, AKPGDGAGSGPR and GLPGLGPAGPKNot specified
*Scomber scombrus*
Antibacterial200 μg/mL[112]N-KVEIVAINDPFIDL-CNot specified
*Scomber scombrus*
AntibacterialMIC 0.263, 0.131, 0.131, 0.263, 0.263 mMagainst *L. acidophilus, L. ivanovi, L. monocytogenes, M. luteus, B. thetaiotaomicron*
[113]GWGSFFKKAAHVGKHVGKAALTHYLSkinWinter flounder (*Pleuronectes americanus*)AntimicrobialMIC 1.1–2.2, 4.4–8.8, 17.7–35.0, 2.2–3.3, 8.8–17.7, 17.7–35.0 μM against *B. subtilis, P. haemolytica, S. aureus, E.coli, S. typhimurium (I and II), A. salmonicida*
[111]GPL, GPMSkin
*Theragra chalcogramma*
AntihypertensiveIC_50_ = 2.6 and 17.1 μM[115]DPALATEPDPMPF Not specified
*Oreochromis niloticus*
Antihypertensive1–20 μg/mL[117]FGASTRGAFramePollack (*Theragra chalcogramma*)AntihypertensiveIC_50_ = 14.7 µM[116]GPEGPAGAR GETGPAGPAGAAGPAGPR Skin
*Oreochromis niloticus*
Antihypertensive5 mg/mL[118]MVGSAPGVL, LGPLGHQSkinSkateAntihypertensiveIC_50_ = 3.09, 4.22 µM[119]GASSGMPG, LAYASkinPacific cod skinAntihypertensiveIC_50_ = 6.9, 14.5 µM[120]Collagen peptidesSkin
*Theragra chalcogramma*
AntihypertensiveIC_50_ = 0.49 mg/mL[122]GIPGAP and APGAPSkin
*Raja clavata*
AntihypertensiveIC_50_ =27.9, 170.2 μM[123]GY, VY, GF, VIYScaleSea BreamAntihypertensiveIC_50_ = 265, 16, 708, 7.5 µM[124]PGPLGLTGP, QLGFLGPRSkinSkateAntihypertensiveIC_50_ = 95, 148 µM[125]GLPLNLPSkinSalmon (*Oncorhynchus keta*)Antihypertensive18.7 µM[126]MIFPGAGGPELFrameSole (*Limanda aspera*)Antihypertensive28.7 μg/ml[127]GDLGKTTTVSNWSPPKYKDTPFrameTunaAntihypertensive11.28 μM[128]Hydrolysates with MW between 1000 and 10,000 DaBoneYellowtailAntihypertensive,antioxidant1.9 mg/mL (ACE inhib.), ~10 mg/mL (DPPH)[129]Hydrolysates with MW ≤ 1000 DaBoneYellowtailAntihypertensive,antioxidant1.5 mg/mL (ACE inhib.), ~35 mg/mL (DPPH)[129]Hydrolysates with weight ~ 30 KDaFrameCodAntioxidant,antihypertensive~20% (antiox.), ~40 mg protein/mL × 100[130]Hydrolysates with weight ~ 10 KDaFrameCodAntioxidant,antihypertensive~15% (antiox.), ~35 mg protein/mL × 100[130]Hydrolysates with weight ~ 5 KDaFrameCodAntioxidant,antihypertensive~40% (antiox.), ~20 mg protein/mL × 100[130]Hydrolysates with weight ~ 3 KDaFrameCodAntioxidant,antihypertensive~18% (antiox.), ~8 mg protein/mL × 100[130]PYSFK, GFGPEL, VGGRP.SkinGrass carp (*Ctenopharyngodon idella*)Antioxidant DPPH radical 2.459, 3.634, 6.063 mM (DPPH), 3.563, 2.606, 4.241 mM (hydroxyl), 0.281, 0.530, 0.960 mM (ABTS)[132]AVGATSkinThornback rayAntioxidant33% of activity at 3 mg/mL (DPPH)[123]DPALATEPDMPFSkinNile tilapia (*Oreochromis niloticus*)Antioxidant8.82 µM (DPPH), 7.56 µM (Hydroxyl)[133]PFGPD,PYGAKG,YGPMSkinSpanish mackerel*Scomberomorous niphonius*Antioxidant0.80, 3.02, 0.72 mg/mL (DPPH), 0.81, 0.66, 0.88 mg/mL (hydroxyl), 0.91, 0.80, 0.73(superoxide anion), 0.86, 1.07, 0.82 mg/mL (ABTS)[134]GSGGL, GPGGFI, FIGPSkinBlue leatherjacket (*Navodon septentrionalis*)Antioxidant405 µg/mL (DPPH), 179 µg/mL (Hydroxyl); 194 µg/mL (DPPH), 89 µg/mL (Hydroxyl); 118 µg/mL (DPPH), 73 µg/mL (Hydroxyl)[135]GLFGPR GATGPQGPLGPR, VLGPF, QLGLGPVSkinSeabass (*Lates calcarifer*)Antioxidant81.41, 10.4, 2.59, 0.50 mmol Trolox equivalents/µmol peptide (ABTS)[136]FDSGPAGVL, DGPLQAGQPGERSkinJumbo squid (*Dosidicus gigas*)Antioxidant_[137]PAGTSkinAmur sturgeonAntioxidant5380 µg/mL (DPPH), 890 µg/mL (Hydroxyl), 8 µg/mL (ABTS)[138]EGL, YGDEYSkinNile tilapia (*Oreochromis niloticus*)Antioxidant4.61 µg/mL (Hydroxyl), 6.45 µg/mL (Hydroxyl)[134]LSGYGPSkinTilapia (*Oreochromis niloticus*)Antioxidant22.47 μg/mL[140]HydrolysatesSkinSnapper (*Priacanthus macracanthus*)Antioxidant_[141]HydrolysatesSkinSnapper (*Lutjanus vitta*)Antioxidant_[142]HydrolysatesSkinSoleAntioxidant_[143]HydrolysatesWB, Head, GonadsHerring (*Clupea harengus*)Antioxidant_[144]TCSP, TGGGNVSkinCod (*Gadus microcephalus*)Antihypertensive, Antioxidant81%, 68% at 500 µg/mL, 75% at 500 µg/mL (scavenging activity) for both[145]N-terminal RPDFDLEPPYFrameSole (*Limanda aspera*)Antioxidant_[147]Collagen/gelatin/peptides Skin
*Thunnus albacares*
Antioxidant9–700 μg/mL[92]GLFGPR Skin
*Lates calcarifer*
Antioxidant5, 10 mg/mL[152]HGPLGPLSkinHoki (*Johnius belengerii*)Antioxidant156.2 µM (DPPH)[137]GPRGTIGLVG, GPAGPAG and GFPSGScales
*Pseudosciaena crocea*
AntioxidantIC_50_ (mg/mL): 0.293, 0.240, 0.107, (hydroxyl); 1.271, 0.675, 0.283 (DPPH); 0.463, 0.099, 0.151 (superoxide anion); 0.421, 0.309, 0.210 (ABTS).[153]HGPHGE, DGPKGH and MLGPFGPSScales
*Katsuwonus pelamis*
AntioxidantEC_50_ mg/mL: 1.34, 0.54, 0.67 (DPPH) 1.03, 0.41, 0.74 (hydroxyl) 1.19, 0.71, 1.59 (superoxide anion) [100]GAEGFIFBone
*Katsuwonus pelamis*
AntioxidantEC_50_ mg/mL: 0.57, 0.30 (DPPH); 0.25, 0.32 (hydroxyl) 0.52, 0.48 (superoxide anion) 0.41, 0.21 (ABTS) [154]GPE, GARGPQ and GFTGPPGNG Cartilage
*Sphyrna lewini*
AntioxidantEC_50_ mg/mL: 2.43, 2.66, 1.99 (DPPH); 0.28, 0.21,0.15 (hydroxyl) 0.24, 0.18, 0.29 (ABTS); 0.10, 0.14, 0.11 (superoxide anion) [155]YGCC, DSSCSG, NNAEYYK and PAGNVRSkin
*Theragra chalcogramma*
AntioxidantIC_50_ = 7.63 μg/mL[156]NHRYDRSkinHorse Mackerel (*Magalaspis cordyla*)Antioxidant72.3% (DPPH), 51.2% (Hydroxyl)[157]GNRGFACRHASkinCrocker (*Otolithes ruber*)Antioxidant79.6% (DPPH), 56.8% (Hydroxyl)[157]QGYRPLRGPEFLSkinSkate (*Raja kenojei*)Neuroprotective 24.26 μM[158]Collagen peptidesSkinSalmon (*Oncorhynchus keta*)Neuroprotective 0.33, 1.0, 3.0 g/kg rat body[159]Collagen peptidesSkinSalmon (*Oncorhynchus keta*)Neuroprotective 0.22%, 0.44% or 1.32% wt/wt diet[160]Collagen peptide drinkNot specifiedNot specifiedAntioxidant0.25% weight/volume[162]Hydrolyzed collagenNot specified
*Pangasius hypophthalmus*
Skin elasticity10 g daily[161]EIGPSGGRGKPGKDGDAGPK, GFSGLDGAKGDSkinCodMatrix metalloproteinase inhibitory activity0.1 mg/mL[163]Collagen tripeptideSkinSutchi catfish (*Pangasius hypophthalmus*)Matrix metalloproteinase inhibitory, anti-photoaging 167–333 mg/kg/day [164]Collagen hydrolysateNot specifiedNot specifiedOsteoclastic differentiation of BMSCs0.2 mg/mL[165]SWFCPSkinTunaAdipocite differentiation0.5–1 mg/mL[166]FIMGLYCartilage
*Raja porosa*
AnticancerIC_50_ = 4.81 mg/mL[168]Collagen hydrolysateSkin
*Aluterus monoceros*
Anticancer/antidiabetic/wound healing0.05–1 mg/mL[169]Collagen peptidesSkin
*Oncorhynchus keta*
Bone regeneration1.125, 2.25 or 4.5 g kg^−1^ BW[170]Collagen peptideSkinGadiformesspeciesChondroprotective1 g/day[171]VLSGGTTMYASLYAEFrameHoki *(Johnius belengerii*)Calcium binding-[172]VLSGGTTMAMYTLVFramePollack (*Theragra chalcogramma*)Calcium binding-[173]Phosphopeptide (FBP)BoneHoki (*Johnius belengerii*)Calcium binding-[174]GPAGPHGPPGKDGR, AGPHGPPGKDGR, AGPAGPAGARSkinPacific codIron-chelating_[175]


### 3.3. Chitin

An analysis of the business implications due to the Covid-19 crisis led to resizing the parameters of the global market for chitin and chitosan derivatives. It was estimated at 106.9 thousand metric tons in 2020 [176], and it is expected now to reach a revised size of 281.7 thousand metric tons by 2027 with an increase at a CAGR of 14.8% in the period 2020–2027 [177]. Before the Covid-19 pandemic occurrence, chitosan global market was expected to increase at a CAGR of 24.7% from 2020 to 2027, while it is estimated now to reach a 16.9% CAGR and 173.9 thousand metric tons by the end of the analysis period. Chitosan is used in large quantities in the production of several consumer products, i.e., antiseptics, food items, cosmetics, medicines, and textiles, and is very attractive for its numerous biological properties and as a therapeutic agent because of its antibacterial and antifungal characteristics. It was reported that the annual production of chitosan accounted for 2000 tons, with the main sources being shrimp and crab shell residues [178]. China should achieve a market size of 48.5 thousand metric tons by the year 2027 with a CAGR of 14.1% in the period 2020 to 2027, while in Europe, Germany is expected to reach a 10.6% CAGR.

Chitin is a long chain odourless/tasteless amino polysaccharide of white or off-white colour in its pure state, composed of *N*-acetyl-*β*-D-glucosamine units and monomers attached via *β* (1→4) linkages [179]. It is considered the second most abundant high MW natural biopolymer after cellulose, to which it is strongly similar in structure. The sole difference lies in the substitution of the secondary hydroxyl groups with acetamide groups on the alpha carbon atom of the cellulose molecule [180,181]. An important chitin derivative is the chitosan, obtained by deacetylation of chitin, and with a molecular structure in which the 2-amine-2-deoxy-D-glucopyranose units predominate [182]. Differently from most of the natural polysaccharides that are of neutral or acidic nature, chitin and chitosan are highly basic [183,184].

On the basis of the different arrangement of chains in the crystalline regions, chitin has been classified in three forms, namely the α-, β-, and γ-forms [10]. Each of them provides to the chitin specific properties, which could be affected also by the different degrees of deacetylation. All these structural combinations led to a wide range of possible activities of interest for several biotechnological industries and tissue engineering. According to [185], it is really hard to achieve the suitable purity degree due to some compounds that naturally occur within chitin, for which the complete elimination is difficult. Indeed, chitin is often included in a complex matrix including proteins and calcium carbonate, with the establishment of strict interactions. Thus, the common procedure employed for the extraction of chitin requires the removal of these associated components, and follows two sequential chemical steps: deproteinization, for the removal of proteins; demineralization, aimed at removing the inorganic component. In some cases an additional step of decolorization/deodorization is applied to remove pigments [186,187]. The experimental extraction procedure is a very delicate and determinant phase for the final application, because purity and crystallinity levels depend on it [188]. 

Until now, the main source of chitin and derivatives were members of the phylum Arthropoda, including crustaceans and insects, provided of an exoskeleton predominantly constituted of chitin [10,189,190] and therein references. Recently, fish wastes have acquired also great attention as potential sources of chitin and derivatives, with particular regard for fish scales, but the use of these kinds of sources is less investigated. Taking into account that the most commonly used resources for the extraction of chitin and chitosan are crustaceans, many approaches focused on the use of fish wastes have employed the same general procedure, with slight variations in some cases. The numerous efforts performed in the isolation of chitin and chitosan from crustaceans reported generally a final yield of chitin for an amount ranged from 14% to 25%, but the use of fish scales seems to be equally promising.

The available literature includes studies investigating the use of scales obtained from four fish families, including the carp, tilapia, red snapper, and parrotfish (Table 3). The first attempt was carried out by Zaku and co-authors [191], which used scales of common carp fish *Cyprinus carpio* to isolate chitin. The fish scales were treated with maceration procedure after drying for three days, washed from residual minerals with acid (HCl 1M) at 30 °C and finally rinsed with deionized water and deproteinized at 95 °C. The yield of chitin expressed as percentage of the starting raw materials was 20.49%, thus comparable with results obtained from crustaceans.

Generally, the physicochemical properties of chitin are investigated through techniques including IR spectroscopy and scanning electron microscopy (SEM), but in some cases the approaches included more specific techniques. This is the cases reported by Kumari et al. [192,193], which obtained and characterized chitin and chitosan from scales of *Labeo rohita* by using X-ray diffraction, elemental analysis, Fourier-transform infrared spectroscopy (FTIR), SEM and differential scanning calorimetry (DSC). For these studies the scales were cleaned, washed, and dried in sunlight for 4 days and then subjected to demineralization and deproteinization. In addition to the studies of Kumari and co-authors, other researchers have also investigated the potential of the *L. rohita* species, even if they have focused more on the purification of chitosan, considered a compound with greater versatility of application. After a procedure including deproteinization and demineralization, the yield of prepared chitin was found to be 22.36%, while for chitosan a final yield of 7.72% was achieved [194]. The degree of deacetylation (DDA) is regarded as an important parameter for the effective application of chitosan, and the recognized good standard for a typical commercial chitosan ranges between 66% and 95% [195]. Muslim et al. [194] which their results, strongly encouraged the possible introduction of chitosan in several application fields, as they detected a %DDA of chitosan of 78.2%. Uawonggul et al. (2002) [196] and Boarin-Alcalde and Graciano-Fonseca [45] managed to obtain chitin and chitosan from scales of Nile tilapia (*Tilapia nilotica*). In particular, Boarin-Alalde and Graciano-Fonseca [45] tried to adjust the procedure generally used to isolate these compounds from crustaceans, including a pretreatment, demineralization, deproteinization, depigmentation and deodorizing and finally deacetylation to obtain chitosan. From 50 gr of raw materials (fish scales) the yield of chitin was the 20%, while the authors achieved a total amount of purified chitosan up to 39%. These percentages are lower than those obtained by crustaceans, but quite in line with other reports on the use of fish scales as source of isolation. More recently interesting results have been obtained from typical tropical saltwater fishes species, namely red snapper and parrotfish. Rumengan et al. [197] obtained a chitin yield of 45% and 33%, from parrotfish (*Chlorurus sordidus*) and red snapper (*Lutjanus argentimaculatus*), respectively, while Takarina and Fanani [198] achieved DDA values up to 75% for chitin and 90.83% for chitosan, by suggesting a huge potential for red snapper fish scales. An even higher value of deacetylation was retrieved in the chitosan obtained from Papuyu fish scales, with 97.40% against the 93.80% of commercial chitosan from shrimp shell [199]. In general, the studies conducted so far are limited to evaluating the possibility of fish scales as a possible alternative source, but are similar in scope and final considerations. The promising results strongly encouraged the exploitation and enhancement of this waste, whose production takes on increasing connotations every year.


marinedrugs-19-00116-t003_Table 3Table 3Chitin/Chitosan from fish waste and their possible applications.CompoundByproductSourceApplicationsReferenceChitinScales
*Cyprinus carpio I.*
Not specified[191]Chitin, ChitosanScales
*Labeo rohita*
Not specified[192]Chitin, ChitosanScales
*Labeo rohita*
Not specified[193]ChitosanScales
*Labeo rohita*
Not specified[194]Chitin, ChitosanScales
*Oreochromis niloticus*
Not specified[45]ChitinScales
*Chlorurus sordidus*
Not specified[197]Chitin Scales
*Lutjanus argentimaculatus*
Not specified[197]Chitin, ChitosanScales*Lutjanus* sp.Not specified[198]ChitosanScales
*Anabas testudineus*
Coagulation-flocculation treatment for iron removal[199]


#### Fish Chitin Applications

The general applications of chitin and chitosan have been extensively reviewed by several authors [10,42,200,201]. The properties possessed by chitin and chitosan, as for example ability to form polyoxysalts and films, to chelate metals and the optical structure characteristics [202] make them attractive compounds for utilization in a number of fields, such as the medical, pharmaceutical, food and cosmetic industries, nutraceuticals, bioremediation, gene therapy and cosmetics. The different approach led to the detection of a more crystalline structure for chitosan than for chitin and fish scale, by confirming its possible use as food supplement, drug preparation and water treatment. Although the results obtained are promising, the available studies that analysed the use of fish scales as a source of chitin and chitosan are limited to the chemical purification and characterization of these compounds, but do not investigate deeper their potential applications. Most of the researchers involved in this field only suggest the possible uses on the base of the chemical characteristics of chitin and chitosan. The only exception is represented by the study of Irawan et al., [201] which demonstrated the promising employment of chitosan in environmental recovery. In the bioremediation field, diverse chitin derivatives possess specific properties useful for treatment of contaminants, as the different functional groups they have could easily interact with heavy metal ions by helping their removal in aqueous solution [190,203]. The chitosan extracted from the fish scales of Papuyu fish has been proved to improve the removal of iron in the groundwater from 11.80 mg/L to 3.43 mg/L, by evidencing higher efficiency in coagulation/flocculation treatment than the commercial chitosan from shrimp shell [199].

Despite the inconsistency of available information, we can obtain some interesting insights from the existing studies referring to molecules extracted mainly from crustaceans, which, thanks to the intense work done in recent decades for their applicability, are considered more fruitful than cellulose [204]. In the attempt of summarize the huge amount of possible applications, notable uses have been reported in the medical and pharmaceutical fields due to the proved biological activities as such as antimicrobial, antioxidant, antitumor immunoadjuvant, antithrombogenic, anticholesteremic and bioadhesivity. Antibacterial activities have been reported for chitosan by several authors [205,206], and were also related to the MW of chitosan by suggesting interaction with cell permeability affecting the inhibition action [207,208]. A mechanism based on the chitosan ability to form permeable films has been proposed at the base of the observed antifungal specific activities [209]. Direct and indirect activity against tumoral cells have been extensively demonstrated for chitosan and its derivatives [210,211,212]. The many possible conformations of chitin and chitosan (fibers, powders, films, sponges, beads, gels, capsules) allow the exploitation of chitosan for drugs administration in several ways and for tissue engineering and wound care dressing [213,214]. Beneficial properties, such as biocompatibility, biodegradability, film-forming capacity and gas and aroma barrier are then at the base of the suggested and proved use for food preservation, packaging, and as colour stabilization agents [215,216,217,218,219]. This is particularly true for chitosan, which exhibited a higher solubility in various acidic solvents than the poor soluble and reactive chitin, and possess antimicrobial activity against many pathogenic microorganisms [220]. Films obtained from chitosan showed flexibility and resistance and an excellent ability to form oxygen barriers, thus ideal for storage of fruits, vegetables, eggs and dairy, cereal, meat, and seafood products.

In any case, theoretically, all these applications can also be envisaged for chitin and chitosan extracted from fish scales, considering the chemical data obtained in terms of net yield and degree of deacetylation. Certainly, further investigations should be carried out to verify the real effectiveness, and to evaluate the different yields and quality levels possible correlated to the species of fish considered, as the different environmental conditions or particular adaptations could affect the composition of the scales themselves.

### 3.4. Oil

The global fish oil market size was valued at $ 1,905.77 million in 2019, and is estimated to reach $ 2844.12 million by 2027 with a CAGR of 5.79% from 2021 to 2027 [221]. The EU produces approximately 120,000 tons of fish oil each year, for which Denmark is the largest producing nation. This production is mainly driven by the great demand of fish oil as ingredient in the aquaculture industry, now consuming as much as 90% of global fish oil supplies [222]. However, there is also good potential for the high end/high value markets, especially from fish byproducts. The increasing global trend of fish products processing will increase the volumes of byproducts. In 2016, the global production of fish oil from byproduct accounted for 26% of the total fish oil production [223]. In both the fishing and aquaculture industries, oil and fats represent a significant fraction of finfish processing waste and the amount of which depends upon the fat content of the specific fish species, the distribution of fat in fish parts, its age, sex, nutritional status, health, and time of year determine the amount of oil/fats [41]. For example, it is well noted that the visceral mass of fish discards has a significant amount of oil or fat apart from proteins [224]. Fish oil is found in the flesh, head, frames, fin, tail, skin and guts of fish in varying quantities. Generally, fish contains 2–30% fat, and about 50% of the body weight is generated as waste during the fish processing operation [225], meaning that there is a great potential for valorisation of this waste, mainly for human consumption or to produce biodiesel. Fish oil contains mainly triglycerides of fatty acids (glycerol combined with three similar or different acid molecules) with variable amounts of phospholipids, glycerol ethers and wax esters. It is considered the most nutritious and most digestible ingredient for farmed fish. The lipid composition in fish is quite different from land animal lipids and vegetable oils due to the large quantity long-chain PUFAs, including eicosapentaenoic acid (EPA, C20:5, n-3) and docosahexaenoic acid (DHA, C22:6, n-3) also known as omega-3. These fatty acids, which cannot be synthesized by human body, cover a wide range of critical functions for human health [226,227,228]. Among the best source of oily (EPA and DHA) fish, there are salmon, herring, mackerel, anchovies, sardines and tuna [229]. Fish are not able to synthetize omega-3 but they need to obtain them from the external, through algae and microalgae or plankton in their diets. A wide range of techniques have been used to extract oil from whole fish or fish waste and reported in several reviews [41,228,230]. The selection of the most suitable method depends on different factors, especially by the nature of the waste and by the final application of the oil, whose biodiesel production and food supplements are the most common [46,231]. The applied technologies range from chemical and enzymatic processes, cooking and pressing to more recent green technologies such as microwave and supercritical fluids. A recent work by Mendez and Concha [46] analysed in depth the different extraction methods applied for the production of omega-3. Summarising the main aspects, wet pressing and chemical extraction by solvents are the most common approaches, although the high pressures/temperature and the residual presence of solvents (which needs to be removed using a surplus of energy) limit their use. Less harsh methods use enzymatic hydrolysis by proteases. Different enzymes can be applied, but several studies demonstrated that the use of alcalases is a more efficient process [232,233]. In general enzymatic hydrolysis is a quick and easily reproducible method; it prevents extreme physical and chemical treatments; compared to chemical hydrolysis, it has the advantage of avoiding generating chemical waste, besides being more easily controlled showing great potential for fish waste valorisation. Recently, Araujo et. al., demonstrated that by increasing the initial concentration of alcalase (Eo from 0.94 AU/1 to 4.68 AU/1)—at a constant initial substrate concentration (So = 25 g of protein per liter)— the degree of hydrolysis values and the oil yield increased, obtaining 430 g of protein hydrolysate, 10 g of collagen and 350 g of oil from 1000 g of fish waste [234]. A disadvantage of enzymatic hydrolysis could be the high price of some enzymes which, most of the time, cannot be recycled. In a view of sustainability, more environmentally friendly approaches are needed. In this context supercritical fluid extraction (SFE) is an emerging extraction technology using solvents, mainly CO_2_, that has many advantages compared with traditional techniques because it uses moderate temperatures, reduces the lipid oxidation during the extraction process, allows a selective extraction of low polar lipid compounds, avoiding the co-extraction of polar impurities such as some inorganic derivatives with heavy metals [235]. Several studies demonstrated that this approach provides the same efficiency of oil extraction from fish waste but in a more sustainable way. The application of SFE from the viscera of African catfish Clarias gariepinus and from common carp C. carpio allowed to obtain a yield of oil which was comparable with the yield extracted using the Soxhlet method [236,237]. Furthermore, another cleaner and greener method for the oil extraction, which includes the use of microwaves, has been applied by [238], for biodiesel production. They showed that the microwave lipid extraction was approximately 50% more efficient compared to conventional solvent lipid extraction (Bligh and Dyer method). In the same study they demonstrated also that microwave-assisted transesterification reaction in the presence of KOH catalyst for 10 min at 65 °C lead to an efficient conversion into biodiesel. The Gas Chromatography–Mass Spectrometry (GC/MS) analysis confirmed the presence of good quantity of palmitoleic acid, palmitic acid, oleic acid and eicosapentaenoic acid, which are essential biodiesel components [238]. Depending on the extraction methodology, the extracted crude oil could contain impurities [239], and requires a purification process to reach quality features that make it acceptable for human consumption [240]. Moreover, although fish oil possesses several health benefits, the presence of highly unsaturated fatty acids results in auto-oxidation of fish oil, so the final application of the oil product should be taken into consideration since the first stage of the processing.

#### Fish Oil Applications

Industrial fish processing operations generate a significant amount of wastes, which contain long-chain fatty acids and that can be utilized in a variety of markets [230] including industrial uses, food, feed, and aquaculture and nutraceutical applications (Table 4). The reason for the great interest in fish oil is that it contains two important PUFAs called EPA and DHA or otherwise called omega-3 fatty acids. The two main PUFA applications are as feed/food supplements and biofuel production. 

The aquaculture sector is the predominant market and requires oils with low levels of oxidation, low levels of contaminants, and consistent quality. The nutraceutical market requires oils low in oxidation and contaminants, but also with high levels of omega-3 fatty acids [241]. The omega-3 fatty acids are very well known to have beneficial bioactivities including prevention of atherosclerosis, arrhythmias, reduced blood pressure, benefit to diabetic patients, protection against manic-depressive illness, reduced symptoms in asthma patients, protection against chronic obstructive pulmonary diseases, alleviating the symptoms of cystic fibrosis, improving survival of cancer patients, reduction in cardiovascular disease and improved learning ability [173,226,227,228,242,243,244,245]. Medical research has provided strong evidence to indicate that a diet high in fish and marine omega-3 is linked to a lowered risk of cognitive decline and Alzheimer’s disease (AD) [246] and some clinical trials suggests that supplementation by *n*–3 PUFA extracted from fish improves cognitive functioning in elderly adults with mild to no cognitive impairment [247].

Considering the recognised benefits of EPA and DHA on human health, alternative ways of supplying dietary omega-3 to the consumers have been explored. One of the approaches is to enrich chicken meat with omega-3 fatty acid derived from sustainable marine sources. To investigate this aspect, Moula Ali added different concentrations of fermentative recovered fish oil (FFO) to broilers’ diet, which resulted in incorporation of EPA + DHA in the animal, observing a consequent reduction in cholesterol (ranging from 9.2 to 16.6% compared with control) and triglyceride (ranging from 1.5 to 3.1% compared with control) concentrations in serum, liver and meat of birds fed with FFO. This demonstrated that oil from fish waste can indirectly benefit human health through chicken meat [248]. Several studies demonstrated also the potential of fish oil as antioxidants. In their study, Sellami et al. [249] extracted oil from the waste liver of three ray species evaluating the extract composition and scavenging activity. The fatty acid profiles exhibited a dominance of unsaturated fatty acids (UFAs) exceeding for all samples 65% of the total fatty acids content. The major n-3 PUFAs were EPA (C20:5) and DHA (C22:6) with contents varying from 3.36 to 5.51% and from 9.07 to 30.50% respectively. Interestingly, they found that these oils contained also carotenoids and phenolic compounds, displaying antioxidant activity comparable to that of olive oil. Oil from fish waste was also successfully applied at microbiological level both as bacterial growth substrate and as antimicrobial. The antioxidant as well as the immunomodulatory properties of the combination of fish (salmon) oil with plant extracts was reported by [250], demonstrating that the synergistic effect lead to an increasing of bioactivity compared with fish oil extract alone.

Oil from waste fish has been used also to induce the production of bacterial enzymes. In particular, it is known that microbial lipase production is highly influenced by medium components like nitrogen sources, carbon sources such as fatty acids, triglycerides and carbohydrates which can stimulate or repress lipase production. Cod liver oil 1.5% added to the growth medium of *Staphylococcus epidermidis* CMST Pi 1 demonstrated to be a suitable triglyceride source to increase lipase production (14.8 U/mL) compared with castor oil, palm oil, and other vegetable oils [251]. On the other side, Inguglia et al. [252] qualitatively characterized and investigated the antimicrobial effects of the fish oil extracted from *S. salar* waste samples derived from Italian fish markets and evaluated the potential use of these compounds for treating pathogen infections. By using GC/MS they showed that the specific fatty acid composition of the salmon waste oils was enriched in MUFAs and PUFAs, with special reference to omega-3, -6, -7, -9 fatty acids. The oleaginous extract was tested against two Gram-positive and Gram-negative, respectively *S. aureus* ATCC 6538 and ATCC 25923 and *P. aeruginosa* ATCC 9027 and ATCC15442 demonstrating an inhibition effect with a MIC of 25 and 12.5%, presumably attributed to their action on cell membrane alteration and destabilisation [253].

Another relevant application of oil from fish waste is the production of eco-friendly fuels, especially biodiesel. Waste oils are potentially advantageous over petroleum and virgin vegetable oil based fuels due to waste utilization, and an overall lowering emissions over the life cycle of fuel production, disposal, lower price (25 cents per gallon for fish oil compared to $1.19 per gallon for diesel fuel), and similar calorific value to petroleum distillates [254]. For these reasons, several studies explored and confirmed the potential of fish waste oil for biodiesel production. In a study conducted by Martins et. al. [255] the physicochemical features of the fish-based biodiesel obtained from tilapia waste oil were checked in accordance with the standard requirements established by Brazilian National Petroleum Agency. They confirmed that the obtained biodiesel is in accordance with the specifications of specific mass, kinematic viscosity, water content, acidity level, flash point and oxidation stability proving the potentiality of using residual oil from tilapia waste as a quality raw material in biodiesel production. The same results were obtained by the pyrolysis at 525 °C of waste fish oil as an animal source of triglycerides, showing that it is possible to obtain biofuels with a good similarity to petroleum-based fuels [256]. Velasquez et. al., showed that oil from viscera of Mexican snook (Centropomus Poeyi), black seabream (Spondyliosoma cantharus), king mackerel (Scomberomorus cavalla) and striped mojarra (Eugerres plumieri) collected in a fish market can be successfully converted into fatty acid methyl ester by enzymatic catalysis [30] and Prakash et. al., compared the performance and emissions of Fish Oil Methyl Ester (FOME) with fossil diesel and demonstrated that FOME could be used as alternative fuel to diesel in stationary diesel engines [32].

A recent application of oil from fish waste was its use as fatliquoring agent in leather processing. As a matter of fact, the physical and mechanical properties of the leather lubricated with the sulphated fish oil fat (fish oil converted into fatliquor by a sulphation process using sulphuric acid) were better than those processed using the commercial fat-liquoring agent [48]. The authors calculated the techno-economic feasibility of this kind of application. Including raw materials, transportation and manpower, the price of the fish oil fatliquor is about 1.98 USD/L, which can save 0.07–0.35 USD/L compared to the commercial fatliquoring agent, suggesting that tanneries would be inclined to employ fish waste oil-based fatliquoring process due to its benefits in terms of environmental safety and cost competitiveness.


marinedrugs-19-00116-t004_Table 4Table 4Oil from fish waste and their possible applications. MIC, minimum inhibitory concentration.CompoundBy-productSourceApplicationsActivityReferenceOilViscera
*Labeo*
*Rohita, Catla catla*
Supplement in animal feedingReduction in cholesterol (9.2 to 16.6%) and in triglyceride (1.5 to 3.1%)[248]Cod liver oilLiver*Cod* fishSupplement in bacterial growth media 14.8 U/mL (lipase production)[251]Omega -3, -6, -7, -9 fatty acidsHead, tissues
*Salmo salar*
AntimicrobialMIC: 25 and 12.5 (%*v/v*)[252]*Tilapia* oilViscera, fins, heads, skin, scales and mixed waste
*Tilapia*
BiodieselNot specified[255]OilViscera
*Centropomus Poeyi, Spondyliosoma cantharus, Scomberomorus cavalla, Eugerres plumieri*
BiodieselNot specified[256]Fish Oil Methyl Ester (FOME)Fish wasteNot specifiedBiofuelNot specified[32]Liver oilRay liver waste
*Dasyatis pastinaca, Dasyatis violacea,*

*Rhinoptera marginata*
AntioxidantIC_50_ 0.92 to 2.1 mg/mL (DPPH)[249]Sulphated fatliquorFish wasteNot specifiedLubricantNot specified[48]


### 3.5. Enzymes

The enzyme market size was around $ 6.3 billion in 2017 and will see growth of around 6.8% in the CAGR through 2024. The expansion of the food and beverage industry due to the growing needs of the population, including the need to improve the flavor, quality and texture of food is leading to continued growth of enzymes market. Furthermore, the growth of this sector is also attributable to the enormous applications that enzymes can have in the detergent industry, increasing the effectiveness of detergents by aiding in stain removal [257].

The bioprospecting of aquatic organisms has led to the discovery of several enzymes with catalytic properties potentially useful for several biotechnological applications. These organisms living in diverse and often hostile environmental conditions, incomparable with the terrestrial habitats, have developed enzymes with unique characteristics [26,258,259], capturing the interest of many researchers. 

The internal organs, including stomach, pancreas, and intestines, are the most important fish byproducts in terms of the number of enzymes found, many of which are cold active, present high catalytic activities also at relatively low concentrations, and stability in a wide range of pH. The complete list of isolated and characterized enzymes from fish have been reviewed by [10,43].

Proteases represent the largest group of enzymes naturally found in fish. These enzymes catalyse the hydrolysis of peptide bonds, through different mechanisms of action [260]. They are referred to as exo-peptidases (or peptidases), when they cleave the terminal amino acid of the polypeptide chain, and as endo-peptidases (or proteinases), when the peptide bonds they cleave are internal. Digestive proteolytic enzymes from fish byproducts belong to four different groups, according to the substrate specificity, classified as aspartic proteases (e.g., pepsin, cathepsin D), serine proteases (e.g., trypsin, chymotrypsin), thiol or cysteine proteases (e.g., calpain, cathepsins B, H, L), and metalloproteases (e.g., collagenases) [258,261]. The most extensively studied proteases that are available in fish include pepsin, trypsin, chymotrypsin, and collagenase. Fish pepsins are generally located within the fish stomach and have greater activity in acidic conditions [262]. However, many fish species secrete at least two different pepsins with different optimal pH, commonly referred to as pepsin I and pepsin II [263,264]. As reviewed by [44], these enzymes have been isolated from the gastric mucosa of several fish species, such as sardine, capelin, cod, salmon, shark, mackerel, orange roughy, tuna, trout, and carp. Trypsin is involved in the hydrolysis of peptide bonds in the carboxyl-terminal ends of lysine and arginine residues [265]; it plays an essential role in the digestion of ingested proteins and is also responsible for the activation of the precursor forms of several other digestive proteinases including chymotrypsin [262]. The use of fish trypsin is increasing enormously, thanks to the unique features of these enzymes including the stability and the high catalytic activity in a wide range of pH and temperature values, including hard conditions (38–70 °C; pH value 8–11) [266]. As reported by [43], fish trypsins have been isolated from sardines, capelin, salmon, cod, bluefish, anchovy, Atlantic croaker, carp, aquacultured tilapia, ray fish, mackerel, threadfin hakeling, red snapper, smooth hound. Compared to trypsin, less work has been reported on fish chymotrypsins, isolated so far from carp, capelin, herring, Atlantic cod, rainbow trout, spiny dogfish, sardine. Collagenases are enzymes capable of hydrolysing the polypeptide backbone of native collagen, without denaturing the protein [267], and are classified as metallocollagenases and serine-collagenases, with different physiological roles. Unlike metallocollagenases, zinc-containing enzymes, which show exclusive specificity for collagen, serine-collagenases have a wide proteolytic activity in addition to collagenolytic activity [258]. Collagenases have been isolated from epithelial, cartilaginous, bony tissues, and digestive tracts of several fish [43].

Besides proteases, another family of enzymes widely found in fish are lipases. These enzymes are glycerol-ester hydrolases, which catalyse the hydrolysis of ester bonds in substrates, such as triglycerides, phospholipids, cholesteryl esters, and vitamin esters [268]. Lipases show specificity in terms of fatty acids, nature of the alcohol, and stereospecificity. Digestive lipases have been isolated and characterized from several fish species, including Atlantic cod (*G. morhua*) pyloric ceca/pancreas, Atlantic salmon (*S. salar*) pancreas, red sea bream (*P. major*) hepatopancreas, rainbow trout (*O. mykiss*) inter-cecal pancreatic tissue and pyloric ceca, oil sardine (*Sardinella longiceps*) hepatopancreas, spiny dogfish (*Squalus acanthius*) pancreas, Nile tilapia (*O. niloticus*) stomach, intestine, grey mullet (Mugil cephalus) viscera [10] and reference within.

Considering the high perishability of the waste, it must be of good quality and in relatively high quantities for commercial extraction of enzymes. Some of the valid processes for the isolation of enzymes from waste are: precipitation by salts and polyacrylic acids, isoelectric solubilization/precipitation, ultrafiltration, pH shift, flocculation and membrane filtration, and overcooled acetone extraction [43,258] and reference within.

#### Fish Enzyme Applications

Today enzymatic methods play a key role in the processes used by modern industries to produce a wide range of products for human consumption. Proteases are one of the most interesting groups of industrial enzymes, widely used for several applications, especially in the food industry and as components of laundry detergents [269,270]; approximately 60% of the total enzyme market involves the use of proteases [271]. Marine organisms, including fish, are an excellent source of enzymes and could contribute to the total number of enzymes available on the world market.

The potential advantages of using enzymes include the development of industrial processes as alternatives to mechanical or chemical treatments which often cause damage to the product and reduce its recovery, ensuring better process control, and low energy requirements and costs. Furthermore, enzymes from fish, being extracted from edible animals, have the added advantage of being safe, so that toxicological tests on raw materials are not necessary [43,258].

Fish proteinases generally have high activity in a large range of pH and temperature values, making them suitable for several industrial applications, such as in the detergent, food, pharmaceutical, and agrochemical industries [272]. In particular, several of these enzymes have been isolated from cold-water fish, then inactivated at relatively low temperatures, which is potentially useful in food applications [273] where it is desirable to inactivate the enzyme with a mild heat treatment. For example, trypsin and alkaline phosphatase purified from cold-water fish shown the temperature optimums about 30 °C lower than the homologues from warm-water fish or mammals [274,275]. 

Cold-active fish pepsins from Atlantic cod (*G. morhua*) and orange roughly (*Hoplostetus atlanticus*) are used for caviar production from the roe of different species. For example, in salmon (*S. salar*), the use of these enzymes facilitates the riddling process, increasing the yield from 70% to 90% [27]. Moreover, it has been shown that cod pepsin could be used in the descaling of fish such as hake and haddock; after a treatment of fish with pepsin in weak acid conditions, the scales could be easily removed with a quick passage through a water jet system [276]. Crude pepsin from Atlantic cod is often used for industrial descaling [277]. A commercial product that includes cold tolerant protease from North Atlantic cod is designated as Penzim [44]. In addition, a cod Uracil-DNA Glycosylase (Cod UNG) from Atlantic cod is marketed by ArcticZymes (http://arcticzymes.com, 5 December 2020); it is a heat-labile enzyme, completely and irreversibly inactivated by moderate heat treatment (55 °C). This enzyme catalyses the hydrolysis of the N-glycosylic bond between uracil and sugar, releasing an apyrimidinic site in uracil-containing single-stranded or double-stranded DNA, therefore it is a good candidate for several molecular biology applications.

Fish gastric proteases are salt-activated as these animals absorb salt-water during feeding. This represents an advantageous feature in several applications, such as fermentations, silage and fish sauce, where significant amounts of salt are used; on the contrary, homologues from mammals are generally inhibited by NaCl [278]. Among the enzymes, pepsin plays an important role in the preparation of silage [277]. For example, pepsins isolated from cod viscera were effectively used for the aqueous phase production of cod viscera silage under acidic conditions [279]. Fish proteases could be also used for skin removal from fish, as an alternative method to mechanical or chemical treatments, which often involve damage or reduction of the recovered product [280].

It has been shown that hydrolysis of proteins used in food industries could increase yields in recovery processes, improve functional properties or improve process methodology. In general, under the conditions that characterize enzyme-catalysed processes, the nutritional value of a protein is preserved better than in traditional acid or alkaline hydrolysis [281,282]. Moreover, autolysis via digestive enzymes of the fish itself is a simple and economical industrial method for preparing fish protein concentrate [283,284], with no enzyme costs involved for the process [258]. Fish sauce prepared by autolysis of fish is a popular condiment due to its distinctive flavor and taste. Enzymatic hydrolysis of dietary proteins offers a rapid and reproducible method for the production of large number of fish bioactive peptides. Compared to microbial proteases that are generally applied, endogenous fish proteases have relatively narrow and unique specificities [283,285], therefore have been also used for the preparation of fish hydrolyzates with several bioactivities (see Section 3.2). In particular, seafood proteases from Atlantic salmon and trypsin from fish pyloric ceca have been used [286,287]. Moreover, intestinal protease extracted from bluefin tuna (*Thunnus thynnus*) was used to hydrolyze hoki (*Johnius belengerii*) bones [174]. In addition, marine collagen was extracted from the fins, scales, skins, bones, head, and swim bladders of bighead carp by using collagenases, pepsin from tuna or trypsin from cod or tuna pyloric caeca [288,289,290,291,292,293]

Lipases isolated from Atlantic cod (*Gadus morhua*) guts have been reported to mainly hydrolyze PUFAs compared to shorter chain fatty acids [294]. Thanks to their high affinity for long-chain fatty acids, specificity for particular fatty acids, and regiospecificity [295], fish lipolytic enzymes can be used in the synthesis of structured lipids. Lipase isolated from sea bass liver was used to defat fish skin, leading to the removal of 84.57% fat from the skin [296].

Digestive lipases from fish could also be used for lipid interesterification reactions, such as the transesterification, with minimal or no byproducts, which generally represent a problem in chemical transformations [297]. Some fish digestive lipases are both stereospecific and enantiospecific, and for this reason they could be used in the synthesis of enantiomerically pure products that can be used as specialty pharmaceuticals or agrochemicals. Furthermore, the use of these lipases could be useful for specialty esters for personal care products and environmentally safe surfactants for applications in detergents or as food emulsifiers [297].

Table 5 lists the most interesting fish enzymes because marketed/used for several applications.

## 4. Conclusions

Fish waste is not only a major environmental problem, but also a huge economic loss. For this reason, a better fish-waste management is needed to overcome these important issues. Therefore, today, the development of a sustainable fish waste management plays a key role, as it is closely related to the Waste Framework Directive, which aims to prevent the generation of waste as much as possible, and to use the waste generated as a resource for reuse, recycling, and recovery. In this way, the use of fish byproducts could contribute to the development of products with high-commercial value, and consequently, to economic growth.

The review comprehensively illustrates the compounds that can be obtained from fish discards and byproducts, highlighting how this waste could become an enormous resource for the production of value-added products (e.g., peptides, proteins, collagen, chitin, oil, and enzymes), with several potential applications. The extraction and purification techniques are mainly based on procedures as acid extraction, enzymatic hydrolysis, and fermentation. The attention in choosing the best extraction technique is generally focused on the final yield and, especially in the case of compounds possibly intended for human consumption, on the preservation of the nutritional value as much as possible. Interestingly, new and more sustainable approaches are in progress, i.e., SFE, which we evidenced as emerging technology for fish oil recovery or the use of green and sustainable deep eutectic solvents. Main extraction methods and further processing technologies reported in the text are summarized in Figure 1.

The range of applications for all compounds deriving from fish discards treatment is very wide, by covering the medical, pharmaceutical, and packaging, food and biofuel production fields.

In consideration of the scale of demand for seafood worldwide, the use of fish discards as a source of high-value compounds is an excellent cost recovery strategy, especially in the historical period we are experiencing. Indeed, this review provided also the opportunity to highlight how the global market forecasts for some of these products (e.g., an increase of global fish oil market size from $ 1905.77 million in 2019 to $ 2844.12 million by 2027) and how these forecasts have been changed following the Covid-19 outbreak. For instance, the chitosan global market before the Covid-19 pandemic occurrence was expected to increase at a CAGR of 24.7% from 2020 to 2027, while this percentage is now reduced to 16.9%. Among several social and economic fields, the pandemic has affected also the current food systems, based on a linear and globalized production and consumption model. This highlights the need of a new economic model focused on the social wellbeing and environmental sustainability at the core of the EU’s economic recovery. The circular economy is now considered a pivotal component of the recovery plan [301], and in this context, herein, we present the possible concretization of this kind of economic model in the field of marine resources and fish products.

## Figures and Tables

**Figure 1 marinedrugs-19-00116-f001:**
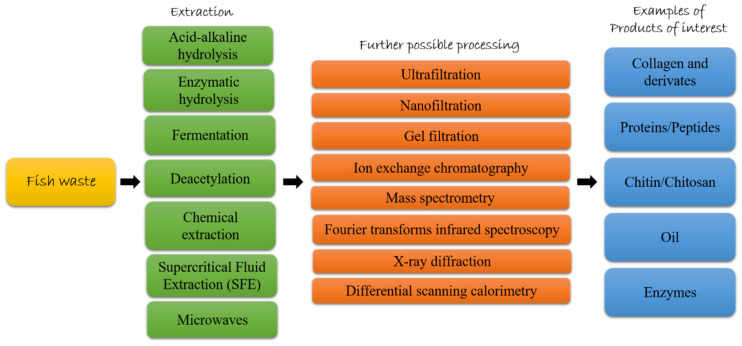
Summary of the main extraction methods and further processing technologies used to obtain high-commercial value compounds from fish byproducts.

**Table 5 marinedrugs-19-00116-t005:** Enzymes from fish waste and their applications. PSC, pepsin-solubilised collagen.

Compound	Byproduct	Source	Applications	Reference
Pepsin	Stomach	Atlantic cod (*Gadus morhua*)	Caviar productionFish descaling	[298]
Pepsin	Viscera	cod	Silage production	[299]
Pepsin	Stomach	Orange roughy (*Hoplostethus atlanticus*)	Caviar production	[299]
Proteases	Pyloric ceca	Atlantic Salmon (*Salmo salar*)	Fish hydrolyzates production	[286]
Trypsin	Pyloric ceca	Unicorn leatherjacket (*Aluterus monoceros*)	Fish hydrolyzates production	[287]
Proteases	Intestine	Bluefin tuna (*Thunnus thynnus*)	Fish hydrolyzates production	[174]
Proteases	Stomach	Albacore tuna (*Thunnus alalunga*)	PSC extraction	[288]
Proteases	Stomach	Yellowfin tuna (*Thunnus albacares*)	PSC extraction	[288]
Uracil-DNA Glycosylase	Liver	Atlantic cod (*Gadus morhua*)	Molecular biology	[300]
Lipases	Intestine/Pyloric ceca	Atlantic cod (*Gadus morhua*)	Potential lipids synthesis	[294]
Lipases	Liver	Sea bass (*Lates calcarifer*)	Defatting of fish skin	[296]

## Data Availability

Not applicable.

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
