# Peer review of "Fish Waste: From Problem to Valuable Resource"

_marinedrugs, 2021, doi:10.3390/md19020116_

Round 1

Reviewer 1 Report

The manuscript is very valuable and provides comprehensive information about health-promoting compounds obtained form fish waste.

I would ask Authort to add two fragments:

First of them would be placed in the section 2 and 3. It would contain brief general information about chemical composition of waste (ranges of protein contents, lipd contents etc.).

The second one would give brief summary concerning technologies applied for processing fish waste. It could be the last section before conclusion. Although manuscript contains information about such technologies in the text, it would be nice for readers to see the summarized for instance in the table.

Author Response

-Reviewer 1

Comments and Suggestions for Authors:

The manuscript is very valuable and provides comprehensive information about health-promoting compounds obtained from fish waste.

I would ask Authors to add two fragments:

First of them would be placed in the section 2 and 3. It would contain brief general information about chemical composition of waste (ranges of protein contents, lipd contents etc.).

We thank the Reviewer for pointing this out. We agree that the composition of waste is an interesting aspect, therefore we included it in the section 3 “Fish by-products as source of high added-value compounds” (lines 166-172) of the new version and added some new references.

The second one would give brief summary concerning technologies applied for processing fish waste. It could be the last section before conclusion. Although manuscript contains information about such technologies in the text, it would be nice for readers to see the summarized for instance in the table.

Authors are grateful to the reviewer for this comment/suggestion. We agree with the referee that it could be nice for readers to include a summary about technologies, but we believe that an additional paragraph before conclusion with this information could be redundant (considering that it is already present in the text). For this reason, we added this summary in the conclusions (lines 990-997; Figure 1) of the revised version.

Reviewer 2 Report

Dear Authors,

The manuscript is a review paper, which generate an overview of fish waste negative impact to us, and detailed discussion of existing researches about fish waste management that could lead to a better future. 

1) A review paper about fish waste as fertilizer in organic farming has published in Waste Management, 2020, 115, 95-112. DOI:10.1016/j.wasman.2020.07.025Authors might considered this utilization as another discussion.

2) Authors might consider the microbial biotransformation of fish waste into industrial methane. Reference: Journal of Cleaner Production, 2021, 279, 123678. DOI: 10.1016/j.jclepro.2020.123678

3) Fish waste also being considered as feeding material for silver pompano fish. I think this is one of the easier method for immediate application. Reference: Animal Feed Science and Technology, 2020, 114748. DOI:10.1016/j.anifeedsci.2020.114748

I felt that the future research direction for fish waste management was not quite clear in conclusion.

Author Response

-Reviewer 2

Comments and Suggestions for Authors:

Dear Authors,

The manuscript is a review paper, which generate an overview of fish waste negative impact to us, and detailed discussion of existing researches about fish waste management that could lead to a better future. 

1) A review paper about fish waste as fertilizer in organic farming has published in Waste Management, 2020, 115, 95-112. DOI:10.1016/j.wasman.2020.07.025 Authors might considered this utilization as another discussion.

We agree with the referee that the use of fish waste as fertilizer is an important aspect, therefore we added this information in the introductory comments (lines 56-57) and in the section 2 “Fish waste in the Circular Bioeconomy Era” (lines 99-102) of the new version. Two reference about this were also added.

2) Authors might consider the microbial biotransformation of fish waste into industrial methane. Reference: Journal of Cleaner Production, 2021, 279, 123678. DOI: 10.1016/j.jclepro.2020.123678

We thank the Reviewer for pointing this out. We added the reference suggested by the referee and the increased production of methane in presence of fish waste, in the section 2 “Fish waste in the Circular Bioeconomy Era” of the revised version (lines 102-104).

3) Fish waste also being considered as feeding material for silver pompano fish. I think this is one of the easier method for immediate application. Reference: Animal Feed Science and Technology, 2020, 114748. DOI:10.1016/j.anifeedsci.2020.114748

We strongly agree with the Reviewer that this is an important aspect to deal with. Therefore, a paragraph was added in the Section 3.2.1 “Fish peptides applications” of the revised version (lines 393-397).

I felt that the future research direction for fish waste management was not quite clear in conclusion.

The final paragraph was revised according to the suggestion of the Reviewer (line 973-979).

Reviewer 3 Report

Fish wastes are sources of important ingredients in the preparation of curative, protective and preventive medicines. Animal-derived medicinal products (ADMP) is part of traditional medicine in different countries, including China, Russia, India and etc.. The use of fish waste modern research methods is an important approach for creating an evidence base for traditional medicine.

ADMP from fish waste preparations are widely used in medicine (https://doi.org/10.1016/j.jep.2019.111933).

The questions of utilization of fish waste were previously considered and discussed in the article (https://doi.org/10.1134/S1063074012060090).

Please note that the possibility of combining fish oil with plant extracts was studied in the article (10.24411/0042-8833-2017-00021, https://doi.org/10.1016/j.synres.2015.07.001 ), while the authors noted the synergism of the resulting combinations (https://doi.org/10.1016/j.synres.2015.07.001).

Author Response

-Reviewer 3

Comments and Suggestions for Authors

Fish wastes are sources of important ingredients in the preparation of curative, protective and preventive medicines. Animal-derived medicinal products (ADMP) is part of traditional medicine in different countries, including China, Russia, India and etc. The use of fish waste modern research methods is an important approach for creating an evidence base for traditional medicine.

ADMP from fish waste preparations are widely used in medicine (https://doi.org/10.1016/j.jep.2019.111933).

We thank the Reviewer for pointing this out. We agree that ADMPs from fish waste used in medicine is important, therefore we added this information in the text of the revised version (lines 161-163).

The questions of utilization of fish waste were previously considered and discussed in the article (https://doi.org/10.1134/S1063074012060090).

The reference was added in the text with respect to peptides extraction methods (line 351) and applications (line 374).

Please note that the possibility of combining fish oil with plant extracts was studied in the article (10.24411/0042-8833-2017-00021, https://doi.org/10.1016/j.synres.2015.07.001 ), while the authors noted the synergism of the resulting combinations (https://doi.org/10.1016/j.synres.2015.07.001).

The paragraph 3.4.1 “Fish oil applications” was revised according to the suggestion of the Reviewer (lines 770; 791-793). The suggested references were also added.

Round 2

Reviewer 3 Report

The authors have made the necessary adjustments, I have no more questions.